



# On the Statistical Relationship between Sea Ice Freeboard and C-Band Microwave Backscatter – A Study with Sentinel-1 and Operation IceBridge

Siqi Liu[1], Shiming Xu[1,2], Wenkai Guo[3], Yanfei Fan[1], Lu Zhou[4], Jack Landy[3], Malin Johansson[3], Weixin Zhu[1], and Alek Petty[5]

[1]Department of Earth System Science, Tsinghua University, Beijing, China
[2]University Cooperation of Polar Research, Beijing, China
[3]UiT – The Arctic University of Norway, Tromsø, Norway
[4]Institute for Marine and Atmospheric Research, Department of Physics, Utrecht University, Utrecht, The Netherlands
[5]Earth System Science Interdisciplinary Center (ESSIC) of the University of Maryland, University of Maryland, College Park, MD, USA

**Correspondence:** Shiming Xu (xusm@tsinghua.edu.cn)

**Abstract.** In this study, we evaluate the statistical relationship between sea ice freeboard and C-band microwave backscatter. By collocating observations between Sentinel-1 images and Operation IceBridge (OIB) measurements in April 2019, we evaluate their relationship under various sea ice types and thickness regimes. We show that, at various spatial scales relevant to synthetic aperture radar (SAR) observations, there exists an apparent significant correlation between C-band backscatter and sea ice freeboard. This relation depends on physical parameters of the sea ice, including the ice type, as well as sensor-specific parameters such as the observational incidence angle of the SAR satellite. As a result, there is considerable variability in this apparent relationship and its fitted parameters. Using the fitted relationship, two-dimensional freeboard maps can be predicted at the scale of SAR images' effective resolution (i.e., $\sim 200m$). More importantly, we demonstrate that although the resolution of SAR images are relatively lower than OIB freeboard maps, we can predict the high-resolution, meter-scale freeboard distribution where altimetry measurements are not available. Thus the representation of altimetric measurements can be improved with the upscaling based on the SAR image. The proposed method can be further utilized for the upscaling of satellite based sea ice topography measurements by the Ice, Cloud, and land Elevation Satellite-2 (ICESat-2). Related issues, including the limitation to spring data, scale dependency and the locality of the statistical relationship, as well as the upscaling of current and historical satellite campaigns, are further discussed.



## 1 Introduction

**Remote Sensing of Sea Ice:** Polar sea ice has undergone drastic changes in response to global climate change (Kwok, 2018). As Arctic sea ice coverage diminishes at a substantial rate, there has also been a rapid decrease in ice thickness and volume (Sumata et al., 2023). In particular, sea ice topography, characterized by the small-scale sea ice height variability, has become smoother

(Krumpen et al., 2025). Satellite altimetry serves as the backbone for observations of the circumpolar sea ice freeboard and thickness. For both laser and radar altimeters, the signals are sent from the satellites to Earth. By measuring the time difference between the emitted pulse from the satellite and the returned echo, the range between the satellite and the reflecting surface on Earth is estimated. The differentiation of the range of echoes returned from sea ice floes versus interstitial leads gives the radar or laser freeboard, and the sea ice thickness is then calculated from hydrostatic assumptions and the buoyancy relationship. In

particular, NASA's ICESat-2 (IS2) satellite is a photon-counting laser altimeter that has carried out continuous observations in both polar regions since 2018 (Markus et al., 2017). Six laser beams of IS2 form into three strong-weak pairs, providing continuous ground coverage in the satellite's flight direction. Validation efforts with airborne campaigns that collocate with IS2 beam segments, including NASA's Operation IceBridge (MacGregor et al., 2021, OIB) and MOSAiC (Nicolaus et al., 2022), show that IS2 is able to achieve highly accurate measurements of the sea ice topography (Kwok et al., 2019; Ricker et al.,

30    2023).

**Problems:** Despite their advantages, satellite altimeters have limited coverage over the sea ice cover. The spatial sampling is inherently confined within the nadir of the satellite's track. For example, the three IS2 beam pairs are within $\sim 3km$ of its ground track. In order to attain basin-scale coverage, samples collected throughout the whole month are usually needed. However, within a month's time, the sea ice may have undergone significant changes due to both thermodynamic and dynamic

processes. These changes cannot be represented by the aggregated monthly freeboard and thickness maps. Furthermore, the altimetric scans only cover limited area within typical passive microwave imagers' footprints, thus hindering the synergy with these observations (Xu et al., 2017). For example, L-band passive microwave radiometer such as the one onboard the Soil Moisture and Ocean Salinity (SMOS) satellite have complementary observational capabilities to altimeters, and they can be physically synergized for the simultaneous retrieval of sea ice thickness and snow depth (Xu et al., 2017; Zhou et al., 2018;

Ricker et al., 2017). However, compared with SMOS's daily basin coverage, much longer periods are needed to obtain an overlapping wide geographic coverage from altimeters such as IS2. Also, small-scale features such as sea ice (refrozen) leads greatly modulate the L-band brightness temperature (TB, see Zhou et al., 2017), but they are potentially not sampled by line scans of altimeters. For example, previous studies (e.g., Fig. A2 of Zhou et al., 2018) show that a remarkable reduction of the TB uncertainty can be achieved with better coverage of freeboard measurements within the SMOS's footprint.

**Paper Info.:** In this paper we explore the potential of improving the laser altimeter's representation through a synergy with microwave backscatter measurements by synthetic aperture radars (SAR). In particular, the C-band SAR payloads onboard European Space Agency's (ESA's) Sentinel-1 (S1) satellites provide pan-Arctic coverage since 2014 through the Extra-Wide (EW) swath mode scans. In this study, we establish statistical relationships between OIB-based sea ice topographic and freeboard measurements and SAR backscatter normalized radar cross section ($\sigma_0$) from S1 scenes using collocated observations





during April, 2019. OIB flights during this month, in particular the Airborne Topographic Mapper (ATM) measurements, were intentionally collocated with IS2 tracks. The ATM measurements feature higher resolution and wider swaths than IS2 measurements, enabling the analysis of co-variability between freeboard and ($\sigma_0$) at multiple scales. Therefore, they are used to study the upscaling of IS2 measurements. In Section 2 we introduce details of the data used and the processing protocols, and Section 3 covers the statistical analysis under various sea ice conditions. Using these statistical relationships, we further

design an algorithm prototype for SAR-based prediction and upscaling of laser altimetry. The locality and limitations of the prediction algorithm are also investigated, along with other related issues in Section 4. Finally, Section 5 includes a summary and the outlook to future work.

## 2    Data and protocols

### 2.1    OIB campaigns in April, 2019

During April 2019 four OIB campaigns were carried out in the Arctic (Fig. 1), which were collocated with IS2 and consequently provided validation data for the sea ice elevation (ATL07, see also: Kwok et al., 2019) and freeboard products (ATL10). In particular, the flights on April 8th and 12th were organized into racetracks and cover more than $200km$ along the corresponding IS2 ground tracks, with outbound (i.e., northbound) and inbound (i.e., southbound) flight passes covering beam pair of #3-#4 and #1-#2, respectively. Two different types of conic scans of ATM onboard these OIB campaigns were carried out: the $15°$

wide swath scan that covers about $500m$ across the flight pass, and the $2.5°$ narrow swath scan that covers about $80m$. The scan angle of the wide-swath scanners is $15°$, resulting in a swath width of $500m$. While the scan angle of the narrow-swath scanners is $2.5°$, which enhances the shot density within the center of the wide swath. In addition, there are three flight passes of the racetrack, and together they cover over $1km$ in the cross-track/flight path direction. Furthermore, the campaign on April 8th dominantly covered areas with thick multi-year ice (MYI), while that on April 12th sampled more interstitial first-year ice

(FYI) within the MYI. Two other flights on April 19th and 22nd are longer tracks that traverse both MYI and FYI (Fig. 1).

In order to fully utilize the ATM measurements on April 8th and 12th, we construct a merged sea ice freeboard map using all three OIB passes. Full details of the processing are covered in Appendix A. Briefly, first, we retrieve the total freeboard (denoted $F_s$) within the entire ATM swath for each pass, using the raw elevation measurements by ATM. Second, we obtain the $1m$-scale $F_s$ map for each pass through spatial linear interpolation. The scan pattern of the ATM results in a variable number

of shot spacings within the scan swath, with lower shot density in the middle (Petty et al., 2016). To mitigate errors introduced by this spatial sampling non-uniformity, the irregularly spaced ATM elevation data are converted to a regularly spaced $1m$ resolution. Finally, the $F_s$ maps of the three passes are stitched together after collocation, producing the $F_s$ map that covers $\sim1500m$ in the cross-flight direction.

The newly constructed $1m$-scale $F_s$ maps are validated with the standard OIB Level4 (L4) product. Specifically, we coarsen

the $F_s$ map to match the $40m$ resolution and the location (nadir to the flight) of the L4 product. Validations show strong agreement, with RMSE of $0.15m$ on April 8th and $0.1m$ on April 12th at $40m$ scale. At $400m$ scale, RMSE further decreased



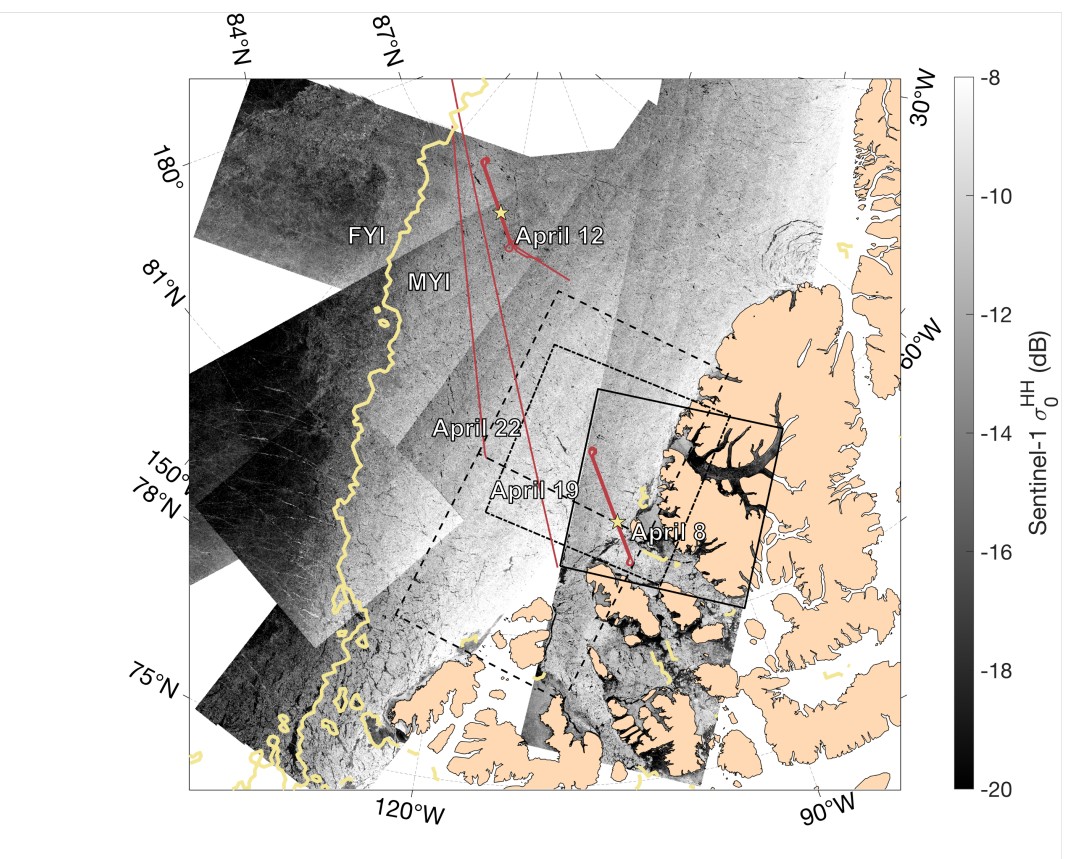

**Figure 1.** OIB campaigns during April 2019. S1 EW images collected around April 8th are shown in the background, with the black boxes outlining the images used for statistical analysis between C-band backscatter and sea ice freeboard. The solid box marks the boundary of the S1 image on April 8th, while the dashed (dot-dashed) ones mark those on April 7th (9th). The OIB ground tracks of the 4 days are marked by red lines, and the location of the sample segments are shown by the asterisks. The thick yellow line delineates the boundary between the MYI and the FYI regions according to the sea ice type product provided by the Ocean and Sea Ice Satellite Application Facility (OSI-SAF).

to $0.04m$ on April 8th and $0.03m$ on April 12th (Fig. S1). Hence the $1m$-scale $F_s$ maps are used further for the statistical analysis with SAR images.

## 2.2 S1 EW images and sea ice type maps

Both S1A and S1B data are available during the study period of April 2019. EW mode images with dual polarization channels (HH and HV) are accessed and collocated with the aforementioned OIB observations. The SAR incidence angles (IA) across the swath range from $20°$ to $46°$ for S1's EW mode. Details of the SAR images, including the image identifiers and the acquisition times, are provided in Tab. B1. Each image is preprocessed using ESA's Sentinel Application Platform (SNAP,





version 11.0.0). Processing steps include the application of precise orbit files, thermal noise correction, radiometric calibration, and terrain correction. Finally, we convert the backscatter intensities into $\sigma_0$.

Sea ice type information is derived from S1 images using a classifier specifically accommodating per-class IA dependencies of SAR intensities (HH and HV) and gray-level co-occurrence matrix (GLCM) textures (Lohse et al., 2020; Guo et al., 2023). Details of this classifier are discussed in Appendix B. Sea ice classification is carried out for all the S1 images and the results are used for further analysis.

By default, the S1 images are projected to $40m$ spatial resolution, which is the nominal pixel spacing of the S1 EW medium GRDM mode data, though the effective resolution is approximately $90m$. In addition, the processing steps in SNAP may further degrade the resolution of the $\sigma_0$ map. This is because a Single Product Speckle Filter with a sliding window of 7×7 pixels were applied during the speckle filtering process. We use the following notations for the coarsened values: $\overline{F_s}^{(s)}$ and $\overline{\sigma_0}^{(s)}$, where $s$ denotes the coarsening scale.

## 2.3 ICESat2 products

The official IS2 products (version 6) are accessed for the collocating tracks with OIB campaigns on April 8th and 12th (see Data Availability for details). Each of the beam segments are of about 150 aggregated photons, and the mean sea ice elevation of each segment is provided in ATL07. Due to the variable photon rates over the sea ice, the along-track length of the beam segment is not constant, around $10-16m$. It is also different between strong and weak beams, with the beam segment length of the weak beams at about $50m$. In this study, we use the footprints of both the strong and weak beam segments to study practical issues limiting the upscaling of IS2 measurements, extending our analysis from OIB to lower freeboard resolution but larger coverage.

## 2.4 Ancillary datasets

The climate data record of global sea ice drift from the Ocean and Sea Ice Satellite Application Facility (OSI-SAF, version OSI-455) is used for the collocation of the different datasets. The OSI-455 product is available for the period of 1991–2020, and is derived from various passive microwave sensors (SSM/I, SSMIS, AMSR-E, and AMSR2) and wind field data from a numerical weather prediction (NWP) model. The sea ice drift vectors are provided on the Equal-Area Scalable Earth (EASE) grid with the spatial resolution of $75km$. However, they are not available near the shoreline (i.e., part of the campaign on April 8th near the Canadian Arctic Archipelago). The temporal scale of the drift vectors is 24-hour, starting/ending at 12:00 UTC (Lavergne and Down, 2023).

## 2.5 Collocation between OIB and S1 images

The collocation between the $F_s$ maps and $\sigma_0$ in the HH-polarization channel is carried out to correct for potential sea ice drift and geocoding uncertainties between the two measurements. For the OIB flight on April 8th, the ice surveyed was relatively immobile, while that covered by the campaign on April 12th experienced a drift of approximately $0.02m/s$ according to the



OSI-455 product. We coarsen the $1m$-scale $F_s$ maps to the nominal pixel size of S1 EW images (i.e., $40m$), and maximize
the correlation (Pearson's $r$) between the two fields by locally adjusting the relative location between the two. The increments
of the local adjustments is $20m$ (i.e., half of S1 EW pixel spacing). In order to compare to the drift corrections during the
correlation maximization(see Fig. 4.a and Fig. 5.a), the daily OSI-SAF drift vectors are scaled to the time interval between the
acquisition time of the SAR image and that of the OIB. Afterwards, bilinear interpolation is carried out in the spatial domain
to attain the drift vector at each location along the OIB flight path.

## 3    Results and analysis

### 3.1    Sample segments

We first examine two OIB segments and collocate the SAR images ($\sigma_0$ in HH-polarization), their locations are shown in Figure
1. For the segment on April 8th, the mean $F_s$ was $1.0m$ with a standard deviation of $0.45m$, and the mean $\sigma_0$ was -10.46 dB
with a standard deviation of 2.77 dB. In contrast, the segment on April 12th had a mean $F_s$ of $0.57m$ and a standard deviation
of $0.18m$, with a mean $\sigma_0$ of -12.67 dB and a standard deviation of 1.52 dB. While the segment covered on April 8th mainly
consisted of thick MYI, that on April 12th features relatively thinner MYI, mixed with FYI and young ice.

The details of the two segments are introduced below.

### 3.1.1    Sample segment on April 8th

The first sample segment is shown in Figure 2. The three OIB outbound flight passes are separated by about 75 minutes: 2019-
Apr-8 12:34 (middle pass), 2019-Apr-8 13:48 (left pass), and 2019-Apr-8 15:01 (right pass), respectively. The inbound flight
passes are: 2019-Apr-8 13:21 (middle pass), 2019-Apr-8 14:34 (left pass), and 2019-Apr-8 15:46 (right pass), respectively.
For both the outbound and the inbound passes, the central pass overlaps with the left (or right) pass by approximately $100m$
in the cross-path direction. The collocation between the passes indicates minimum correction ($1\sim2m$), very high correlations
(Pearson's $r$ over 0.95) and a decorrelation length of less than $5m$ (Fig. S2).

For comparison, the collocation between the merged $F_s$ map and the SAR image on the same day (details in Tab. B1) shows
statistically significant but lower correlation coefficients (Fig. 2.b). The decorrelation distance is much longer than that for
$1m$-scale $F_s$ (i.e. Fig. S2), mainly due to that correlation between $F_s$ and $\sigma_0$ is carried out at the scale of $40m$. Besides, the
statistical relationship between $\overline{F_s}$ and $\sigma_0$ in the HV-polarization channel is also significant, although the backscatter is weaker
by more than 5 dB (Fig. S3).

As mentioned earlier, the effective resolution of the backscatter used in this study is greater than $40m$. Therefore, the coarser
spatial scales adopted for the $\sigma_0$ map is also adopted for the computation of $\overline{F_s}$, i.e. $100m$ (Fig. 2, panel e and h) and $200m$
(Fig. 2, panel f and i).

As shown, the variability of $\overline{F_s}$ is drastically attenuated, but statistical relationship between $\overline{F_s}$ and $\sigma_0$ (at original resolution)
sharpens at larger scales. Specifically, for the segment on the outbound (inbound) flight, the Pearson's $r$ increases from 0.61





(0.66) for the correlation with the $40m$-scale $F_s$ to 0.81 (0.84) for that with the $200m$-scale $F_s$. The slope of the linear fit also reduces slightly as the scale increases, in both cases.

### 3.1.2 Sample segment on April 12th

The other two sample segments are from the campaign on April 12th, shown in Figure 3. The major differences from the
sample segments on April 8th (Fig. 2) are as follows: (1) According to the OIB $F_s$ map, the MYI is much thinner; (2) it contains more areal fraction of FYI, and (3) the surrounding sea ice has undergone more evident drift and deformation between the observations by OIB and S1, as indicated by the OSI-455 product.

Although sea ice is generally much thinner ($1m$-scale $F_s$ mostly under $2m$), a statistically significant relationship is also present between $\overline{F_s}$ and $\sigma_0$ (Fig. 3 and S5). For both the outbound and the inbound segments, OIB has attained sufficient
sampling of MYI, but the representation of FYI is not even. Specifically, on the outbound passes, SAR pixels with $\sigma_0^{HH}$ under 18 dB are scarce, and no level FYI is detected in the area sampled by OIB. For the inbound passes, an apparent nonlinear relationship between $\overline{F_s}$ and $\sigma_0$ is observed for FYI, due to the effect of ice with different levels of deformation. LFYI has a consistently low $\overline{F_s}$ around 20 $cm$ but corresponds to $\sigma_0$ that varies over a large (5 dB) range, whereas DFYI has strongly varying $\overline{F_s}$ up to around 1 $m$ over a small (2-3 dB) range of $\sigma_0$. The linear fitting for MYI is comparable to that for all sea
ice types for the inbound flight (lower panels of Fig. 3). At both $100m$- and $200m$-scale, the linear regressions of $\overline{F_s}$ to $\sigma_0$ show lower fitting slopes for MYI than for those based on all samples. The large variability of $F_s$ at 40-m scale is tightened considerably as the scale increases. In comparison, MYI always has much steeper regression lines for the sample case on April 8th across all analyzed scales (Fig. 2). This result, although potentially affected by the accuracy of the sea ice type map, highlights the importance of the sufficient sampling of various sea ice types to ensure their representation in the study of the
relationship.

Interestingly, for MYI which is well observed by both sample segments on April 8th and 12th, the statistical fittings between $\overline{F_s}$ and $\sigma_0$ show large differences. For the sample segment on April 8th, the regressions ($40m$-scale) are steeper at: $\overline{F_s} = 0.139 \cdot \sigma_0 + 2.443$ with Pearson's $r = 0.410$ (outbound) and $\overline{F_s} = 0.126 \cdot \sigma_0 + 2.236$ with the regression's $R = 0.458$ (inbound). In comparison, for that on April 12th, the fitting slopes are shallower by about 50%: $\overline{F_s} = 0.06 \cdot \sigma_0 + 1.338$ with the regression's
$R = 0.281$ (outbound at $40m$-scale) and $\overline{F_s} = 0.051 \cdot \sigma_0 + 1.204$ with the regression's $R = 0.263$ (inbound). Furthermore, the backscatter is binned at 1 dB intervals, and the mean $\overline{F_s}$ value is calculated for each 1 dB $\sigma_0$ bin. After binning the samples to $\sigma_0$, the regression lines (i.e., between the mean values of $\overline{F_s}$ and $\sigma_0$ in the bins) are also flatter on April 12th ($mean(\overline{F_s}) = 0.051 \cdot mean(\sigma_0) + 1.244$) than on April 8th ($mean(\overline{F_s}) = 0.105 \cdot mean(\sigma_0) + 2.123$). The potential causes of the different fittings include both: (1) differences in C-band backscatter sensitivity to macro-scale topography due to different
ice/snow properties of the two regions, and (2) different imaging configurations of the SAR images. Related issues, such as the effect of IA on the statistical relationships are further discussed in Section 4.1.



**Figure 2.** Total freeboard ($F_s$, colored) and the S1 HH backscatter ($\sigma_0$, background) over sample segments on April 8th, 2019 (a). Boundaries between different sea ice types are marked by contour lines, including MYI, level FYI (LFYI) and deformed FYI (DFYI). The sea ice type information is determined using the classifier described in the Appendix B. The ICESat-2 ground tracks of the three strong beams (#1, #3 and #5) are also shown as thin black lines. Two $10\text{-}km$ segments on the outbound (i.e., northbound) and the inbound flights are marked out by the solid and dashed red boxes, respectively. The scatter plots between $\overline{F_s}$ and $\sigma_0$ after collocation for the outbound (inbound) flights are shown in panels d, e and f (g, h and i). Three spatial scales for computing $\overline{F_s}$ from the $1m$-scale $F_s$ maps are adopted: $40m$ (native resolution of S1 EW mode, d and g), $100m$ (e and h), and $200m$ (f and i). In panels d to i, the dots are color coded according to their ice types, with the solid (dashed) lines showing the linear fitting lines of $\overline{F_s} = a \cdot \sigma_0 + b$ for all samples (only MYI pixels) and the fitted parameters. Also shown in each panel are the mean values of $\overline{F_s}$ and the interquartiles after binning with $\sigma_0$ (1 dB per bin).





**Figure 3.** Same as Fig. 2, but for sample segments on April 12th.

.



## 3.2 Statistics of all segments on April 8th and 12th

For each of the OIB segments on April 8th and 12th, we generate a merged $F_s$ map and collocate it with the SAR images on the same day. The statistical correlations are shown in Figure 4 and 5, respectively.

On April 8th, the local corrections for collocating $F_s$ and $\sigma_0$ are all within $40m$ (Fig. 4.a). The OSI-SAF drift product indicates about $100m$ drift within the northern part of the OIB track, although the drift vectors are not significant given the respective product uncertainties. SAR images from the surrounding days (i.e., from April 7th and 9th, images shown in Appendix B) also show little drift in the sea ice pack surveyed by the OIB campaign (details not shown). In addition, we have attained meter-scale corrections for the collocation of OIB passes (see Fig. A1). Given the relatively coarser resolution of the

SAR images, we assume that sea ice drift and deformation can be ignored when collocating $F_s$ and $\sigma_0$. The detected local corrections in Fig. 4.a may not indicate actual sea ice drifts, but may be due to geolocating uncertainties, such as those induced by geometric corrections of the SAR images. The correlation between $\overline{F_s}$ and $\sigma_0$ at $200m$ scale is statistically significant for all segments (Fig. 4, panel b and d). After binning to $\sigma_0$, the correlation coefficients are mostly over 0.9(Fig. 4, panel c and e).

For the OIB campaign on April 12th, statistically significant large-scale sea ice drift are observed in the surveyed region

(see Fig. 5.a). The lengths of the local corrections for collocating $F_s$ and $\sigma_0$ are about $250m$. The corrections are consistent between the local segment pairs on the inbound and the outbound flights, and they also agree with the large-scale drift in terms of both direction (north-east) and magnitude. Therefore, these local corrections correspond to the actual sea ice drift between the visits by the OIB campaign and S1.

After the corrections, the correlation coefficients are higher and statistically significant for all segments ($p = 0.05$ level).

Moreover, the correlation coefficients after binning are mostly over 0.9 (Fig. 5, panels c and e).

In Fig. 6 we show the linear regressions between $\sigma_0$ and $200m$-scale $F_s$ for all segments on April 8th and 12th. The results indicate that with $\sigma_0$ and the regression relationships, we can estimate the $200m$-scale $F_s$ with high statistical confidence (regressions' $R$-values over 0.3 for most $9km$ segments). Furthermore, the regression parameters show significant variability among different segments, indicating the physical relationship between $F_s$ and $\sigma_0$ is locally variable and/or the uncertainties

in co-location vary locally. Despite this variation, the regression parameters from the inbound and outbound tracks are very similar. For $27km$-long segments, however, these parameters are much less variant, although certain variability still exists on different parts of the flight track. Specifically, for the segments on April 8th, the variance of $a(b)$ has decreased by 48.6% (36.5%) when comparing $27km$-long segments to $9km$-long segments. For the segments on April 12th, the variance of $a(b)$ decreased even more significantly, by 76.8% (78.7%). Besides, the regressions' $R$-values are also higher for $27km$-long seg-

ments for the both segments on April 8th and April 12th. This implies that, small-scale inhomogeneity of the sea ice cover or errors in data co-location, which cause large variability of $a$'s and $b$'s in Figure 6, are effectively attenuated at larger scales. The regression relationships in Figure 6 can be further used for the prediction and construction of $200m$-sclae $F_s$ maps based on SAR. In particular, given to the locality of the relationships, the prediction of $F_s$ map should also be carried out adjacent to the collocating observations by SAR and altimetic scans.





**Figure 4.** Statistical relationship between $F_s$ and $\sigma_0$ for OIB segments on April 8th, 2019. The local corrections to maximize the correlation between $\overline{F_s}$ and $\sigma_0$ are shown for all segments with valid data on the outbound flight (blue) and the inbound flight (dark red). The correlation coefficients before and after collocation are shown for the outbound (panel b and c) and the inbound flights (panel d and e) for all segments, together with those after binning. Statistically insignificant correlations are shown by crosses ($\times$) in the lower panels ($p = 0.05$ significance level).



**Figure 5.** Same as Fig. 4, but for OIB segments on April 12th, 2019.



**Figure 6.** The linear regression from $40m$-scale $\sigma_0$ to the $200m$-scale $F_s$ for all segments on April 8th (a, b and c) and April 12th (d, e and f): $\overline{F_s} = a \cdot \sigma_0 + b$. The regression's parameters, including $a$ (panel a and d), $b$ (panel b and e), and the $R$-value (c and f) are shown, respectively. Two segment lengths are adopted: $9km$ and $27km$.



### 3.3 Prediction of $F_s$ distribution with $\sigma_0$ map

Given that the altimetric scans by OIB (and IS2) have a finer resolution than available SAR images, the regression in Section 3.2 is inherently limited in the spatial resolution of the predicted $F_s$. Moreover, although there is a significant correlation between $F_s$ and backscatter, the variability of $F_s$ is considerable, and no single indicator based on backscatter effectively captures this variability. Therefore, we focus on the prediction of meter-scale $F_s$ distribution (i.e., at the full resolution of the altimeter data) with SAR images based on their collocating observations of high-resolution $F_s$ maps and relatively coarser $\sigma_0$ maps. In particular, in Section 3.2 we find that the relationship between $F_s$ and $\sigma_0$ is spatially localized. Therefore, the prediction is based on segments on the OIB's inbound pass and validated with the adjacent segments on the outbound pass.

#### 3.3.1 Study of sample segments

We first study the sample segments in Section 3.1.1 and 3.1.2. Since the backscatter are binned at intervals of 1 dB, and then we perform statistical fittings of the $1m$-scale $F_s$ distribution for each 1 dB $\sigma_0$ bin. The distributions of $F_s$ in typical $\sigma_0$ bins of these two sample segments are shown in Figure 7 and 8, respectively. The sample $F_s$ distributions after binning all show the following characteristics. First, $F_s$ follows a long-tailed, skewed distribution, which is consistent with various findings in existing studies (Xu et al., 2020; Duncan and Farrell, 2022). Second, there is strong heteroskedasticity associated with $F_s$: for larger $\sigma_0$ bins, the mean value of $F_s$ and the variability of $F_s$ are both higher. Third, the $F_s$ distributions are multimodal, especially for $\sigma_0$ bins that contain both FYI and MYI samples (e.g., left panels in Fig. 7 and 8).

To capture the complex shape of the $F_s$ probability density function (PDF, we use the three-component Log-Logistic mixture distribution to fit the sample PDF in each $\sigma_0$ bin. The fitting results (i.e., Fig. 7 and 8) indicate that the different PDF modes are well captured with very low Kolmogorov-Smirnov (K-S) distance to the sample PDF. We further carry out clustering analysis of the various components, based on the modal $F_s$ values and the corresponding $\sigma_0$ (right panels of Fig. 7 and 8). The three clusters indicate continuous changes of the PDF parameter with respect to $\sigma_0$, and they generally show a good correspondence to these altered sea ice types: FYI, thin MYI and thick MYI. For example, for the sample segment on April 8th, there is prominent presence of MYI with $F_s$ of over $3m$ and $\sigma_0$ of over $-5$ dB (Fig. 7). This is captured by a separate Log-Logistic component which we manually categorize as the thick MYI. This could be sea ice of higher age than that of the thinner MYI which is given by the second component. Another example is that components with very small modal values of $F_s$ manifest even at very large $\sigma_0$ bins (Fig. 7 and 8, lower panels). Due to the relatively coarse resolution of S1 images, thin MYI may be present in pixels with otherwise large values of both mean $F_s$ and $\sigma_0$. These components are captured by the PDF fitting, and we further manually categorize them as FYI. It is important to note that these categorizations are introduced to interpret the fitting results, as the specific categories (FYI, thin MYI, and thick MYI) were not previously defined in our analysis. Based on the per-bin $F_s$ fittings on the inbound sample segments, we carry out the prediction of $F_s$ distribution on the corresponding outbound segments. Specifically, based on the observed $\sigma_0$ map on the outbound segment, we: (1) formulate the distribution of $\sigma_0$, (2) compute the $F_s$ distribution according to the sample probability of each of the $\sigma_0$ bin, and (3) construct the overall $F_s$ distribution on the outbound segment. For the sample segments on April 8th, the per-bin Log-Logistic mixture fittings





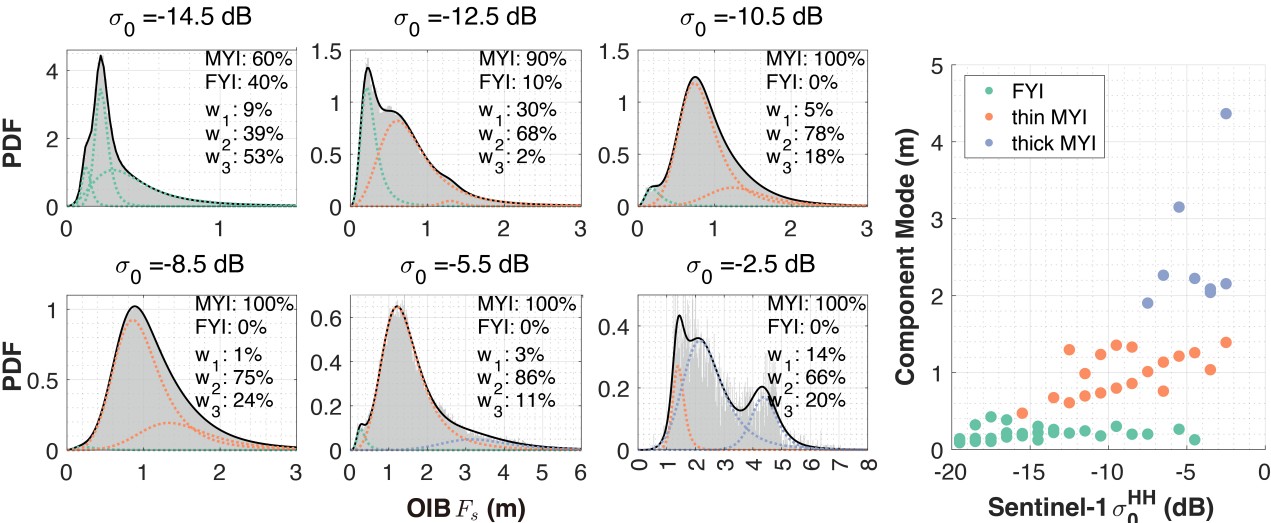

**Figure 7.** Distribution of $1m$-scale $F_s$ in typical $\sigma_0$ bins of the inbound sample segment on April 8th, 2019. $F_s$ sample PDFs, as well as the fitted three Log-Logistic mixture components are shown for typical $\sigma_0$ bins (left panels). Statistical PDF fitting (black solid line) based on the 3-component Log-Logistic mixture model in each panel, along with each of the components (colored dash lines).

demonstrate a high degree of accuracy in fitting the observations for both the inbound and the outbound segments, with K-S distances of 0.002 for each segment. However, the inbound and the outbound segments differ in the sample $F_s$ distribution (Fig.

250 9.b), primarily attributed to variations in the thickness of FYI and MYI, as well as differences in their respective proportions. Notably, the modal thickness values of both the thin MYI and the thick MYI are $0.1m$ higher on the outbound segment than on the inbound segment. As a result, the predicted $F_s$ distribution also shows lower modal $F_s$ values (Fig. 9.a). Despite the underestimation of the modal $F_s$, the prediction is closer to the observation, with lower K-S distance: 0.072, compared with 0.076 between the inbound and the outbound segment.

255 For the sample segments on April 12th, the prediction also shows lower K-S distance with the observed $F_s$ distribution on the outbound flight (K-S distance from 0.094 to 0.074). The major improvement is due to different portions of thin MYI on the outbound and the inbound segments (see also Fig. 3). By using the $\sigma_0$ map on the outbound segment, we achieve the correct representation of thin MYI on the $F_s$ distribution.

### 3.3.2 Validation of prediction for all segments

260 We carry out the prediction of $1m$-scale $F_s$ distribution for all the outbound segments. The validation is based on the K-S distance between the observed $F_s$ sample distribution and the predicted PDF. The baseline is the K-S distance between the observed samples on the inbound and the outbound segments. Figure 10 shows that the predicted $F_s$ PDF is close to the observation, with the mean K-S distance at 0.077. There is a 10% reduction of the baseline K-S distance, which indicates that





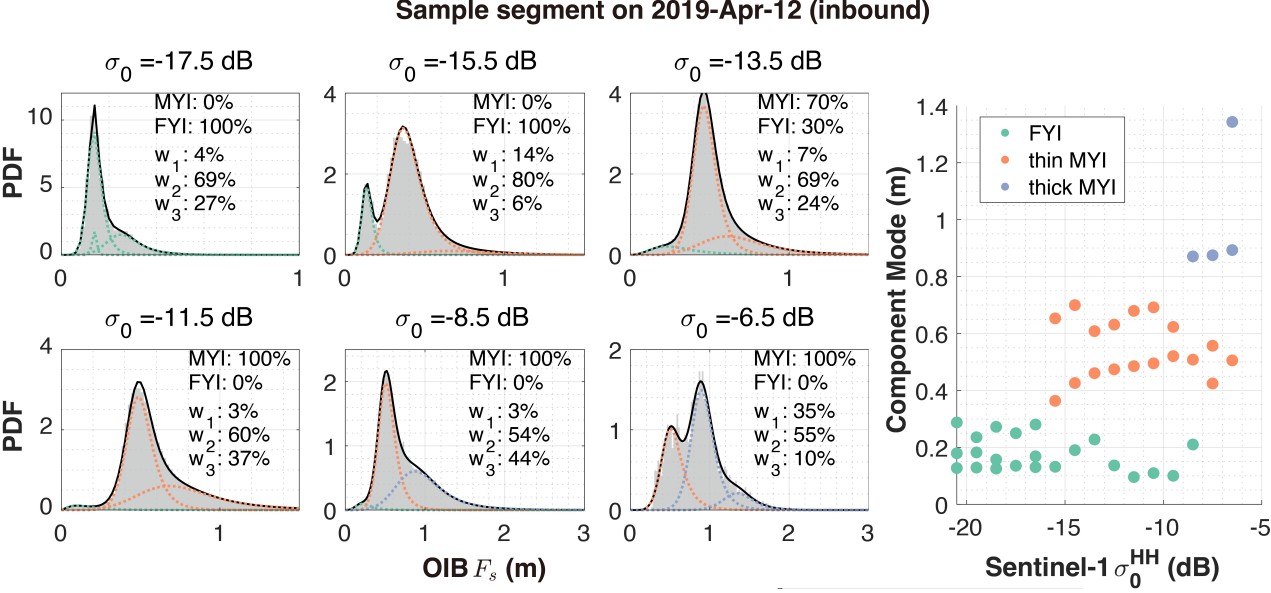

**Figure 8.** Same as Fig. 7, but for the inbound sample segment on April 12th, 2019.

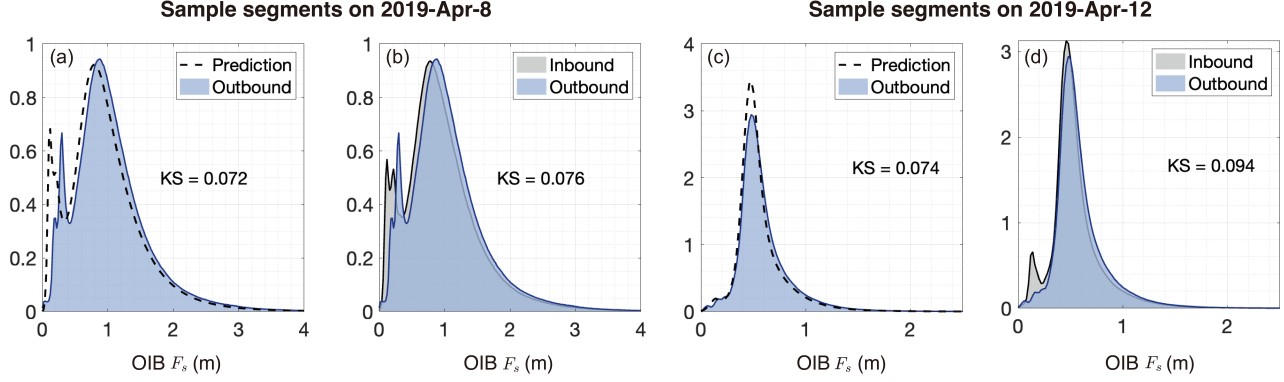

**Figure 9.** Statistical prediction of $F_s$ distributions on the outbound segment with: (1) the per-$\sigma_0$ bin Log-Logistic mixture fittings on the corresponding inbound segment, and (2) the $\sigma_0$ map on the outbound segment. The observed and the predicted $F_s$ distribution, as well as the K-S distance between the two are shown for the sample outbound segment on April 8th (panel a) and April 12th (panel c). The $F_s$ sample distribution on the inbound and the outbound segments are also shown for comparison (b and d).





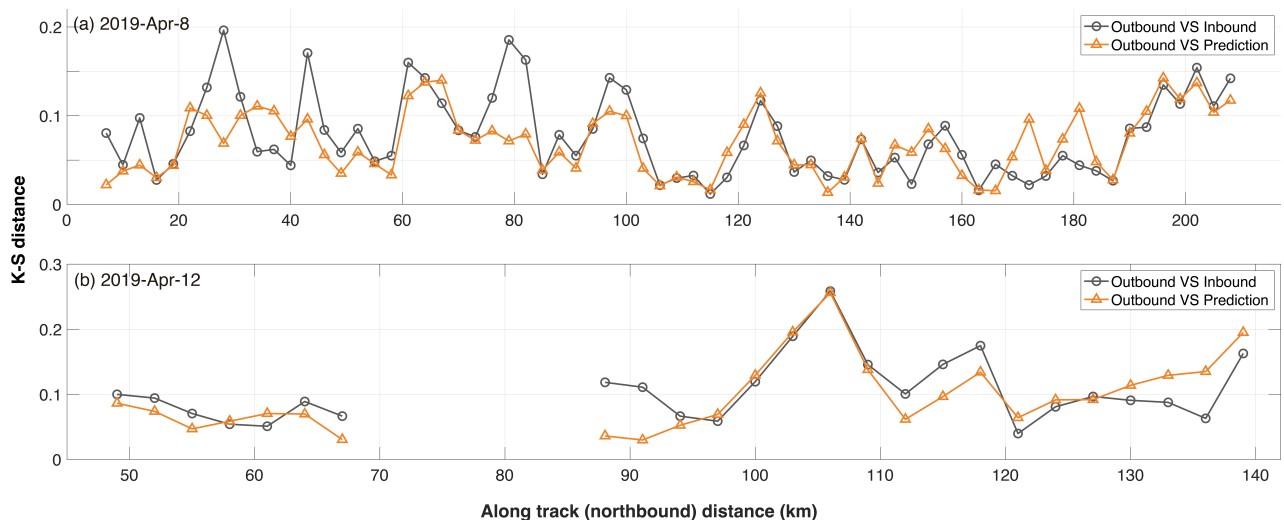

**Figure 10.** K-S distance between the predicted and the sample $F_s$ distribution on all the outbound segments on April 8th (top panel) and 12th (bottom panel). The prediction on each outbound segment is carried out with the PDF fittings on the corresponding inbound segment. The K-S distance between the inbound and the outbound sample $F_s$ distributions are also shown.

the predicted $F_s$ distribution better matches the observations. Especially, large K-S distances are effectively attenuated with the prediction: 3 (10) out of the total 91 segments show a K-S distance over 0.15 between the predicted (inbound) $F_s$ with the outbound observations.

Moreover, there exists a significant positive correlation (Pearson's $r$: 0.72, $p$-value: $2.48 \times 10^{-16}$) between the K-S distance sequences in Figure 10. This indicates that when the $F_s$ elevation is similar between the inbound and the outbound segments, the prediction is generally better. On the contrary, if the $F_s$ distribution is more different between the two segments, the prediction also deteriorates. Therefore, in order to obtain better predictions, the observed $F_s$ should be representative of the sea ice cover on the scale of the prediction. Representation issues for large-scale retrievals are further discussed in Section 4.

## 4 Discussions

### 4.1 Physical mechanisms behind the statistical relationship between $\sigma_0$ and $F_s$

The statistical relationship between sea ice freeboard and C-band microwave backscatter is rooted in the different microwave backscattering mechanisms of various ice surface features. Thin, level ice typically exhibits low backscatter, with two primary scattering mechanisms contributing to this: surface scattering from the ice surface and volume scattering from air voids (Manninen, 1992). However, with thicker ice and larger $F_s$, both the backscatter and $F_s$ variability are higher, as evidenced by the larger spread of $F_s$ interquartiles in higher $\sigma_0$ bins in Fig. 2. This suggests that more complex physical mechanisms govern



the C-band backscatter variations in thicker ice. In the case of older, rougher ice, the presence of thicker snow cover and more
extensive ice deformation leads to increased diffuse reflection and refraction of the incident radar signal (Onstott, 1992).

In addition to the wavelength-scale roughness, several other factors can also influence backscatter, such as the effective radar
incidence angle, radar azimuth which are greatly affected by ridge geometry (Krumpen et al., 2025). For level ice, the effective
incidence angle is relatively constant, equal to the radar incidence angle. However, for ridges, the local incidence angle varies
depending on the radar and ridge geometries, including the incident radar angle, the ridge slope, and the orientation of the
ridge. Even with constant ice properties, these geometric differences alone can lead to higher surface backscatter from ridges
compared to level ice (Manninen, 1992). Consequently, the radar backscatter and its IA dependency is highly dependent on the
ice type and the observational geometry (Geldsetzer and Howell, 2023; Lohse et al., 2021, 2020; Guo et al., 2022).

It is important to note that in this study we did not apply IA corrections to the SAR images. There are several reasons: First,
the IA dependency is type-dependent, with deformed ice showing lower sensitivity to IA than level ice (Makynen et al., 2003).
Given the variant ridge density within the SAR's effective resolution ($\sim 100m$), a simple correction for IA is insufficient in our
study. Second, for the SAR image on April 8th, the IA change was within $10°$ along the whole OIB track, and on April 12th,
IA values were within $5°$. Since the range of IA is small, the correction has potentially limited effect on our study. Third, the
best angle for the IA correction should be chosen to maximize the differentiation among different ice types. What is the best
angle remains an open question and requires more systematic study. We further explore the influence of IA on the statistical
relationship for the OIB track on April 8th (no evident deformation or synoptic event around April 8th). By matching SAR
images from April 7th, 8th, and 9th to the OIB track on April 8th, we obtain the statistical relationships between $F_s$ at different
IAs. In general, the statistical fitting becomes steeper with decreasing IA (Fig. S6). This trend is driven by the higher (lower)
sensitivity of $\sigma_0$ level (ridged) ice to changes in IA (note the weaker $\sigma_0$'s at larger IAs in Fig. S6). Therefore, when IA changes,
the statistically significant relationship still holds, but IA has limited effect on this relationship than other factors, such as the
localized sea ice conditions.

Furthermore, snow cover properties such as snow density and wetness can also modulate the C-band scattering signatures
(Kim et al., 1984). For example, the change in snow density affects the effective wavelength of the microwave signals, therefore
impacting the scattering at the snow-ice interface. Since the OIB campaigns were carried out during later winter/early spring,
the snow cover is dry and therefore largely transparent to C-band signals. In order to apply the statistical prediction algorithm for
other seasons (i.e., late autumn or spring), the snow conditions should be taken into account to better use the SAR measurements
(Livingstone and Drinkwater, 1991).

### 4.2 Scale-dependency of the statistical relationship

Based on the OIB tracks on April 8th and 12th, we further explore the scale-dependent characteristics of the statistical relation-
ship. Specifically, both the OIB $F_s$ and S1 $\sigma_0$ maps are coarsened to three spatial resolutions: $100m$, $200m$ and $500m$. This
coarsening was achieved by calculating the average OIB $F_s$ and S1 intensity within each coarsening grid cell at the respective
resolutions, rather than coarsening the OIB $F_s$ alone as previously shown in Section 3. By analyzing the coarsened $\sigma_0$ and the
coarsened $F_s$ maps, we find that the relationship becomes more stable at large scales (Fig. 11). In several segments, the Pearson





correlation coefficient at $500m$ scale is lower than that at $40m$ and $200m$ scale. This is likely because FYI is distributed across various locations and becomes disappeared after coarsening to the $500m$ scale. On the OIB tracks on April 8th, there is a special segment ( $55km$ in along-track direction) where the Pearson correlation coefficient drops drastically across all three scales. These segments are dominated by deformed and thick ice, with a mean $F_s$ of $1.04m$, a $F_s$ std of $0.56m$, and MYI coverage reaching 97.3%. Moreover, the footprint size of NASA's first ICESat satellite is about $65m$, and the statistical relationship with its concurrent SAR payloads (e.g., ESA's ENVISAT ASAR) can be explored for the prediction of large-scale $F_s$.

Various studies have explored the relationships between sea ice topography and microwave backscatter on different scales, ranging from SAR-related scales (Macdonald et al., 2024; Kortum et al., 2024) to scatterometry scale (Petty et al., 2017). In Macdonald et al. (2024), the Radarsat Constellation Mission (RCM, also C-band SAR) images and ICESat-2 products are used to study the relationship between sea ice roughness and backscatter over land-fast sea ice in the Canadian Arctic Archipelago. In particular, the statistical relationship based on HV polarization is stronger, and therefore used to predict FYI roughness and the height of MYI. In our study, we also find statistically significant relationships on the HV channel (e.g., Fig. S3 and S5). Although the HV-channel usually has a lower signal-noise ratio than the HH-channel, the higher correlations with sea ice topography statistics may arise from the higher dynamic range of $\sigma_0$.

In Kortum et al. (2024) the authors explored the extrapolation of IS2 freeboard (ATL10) with temporally coincident S1 images. Similar to Macdonald et al. (2024), the HV-channel $\sigma_0$ maps are utilized. The prediction is carried out with the pairing CDFs of $F_s$ and $\sigma_0$, and the Pearson correlation coefficient at $400m$ scale reaches 0.82. In our study, the regression model in Section 3.2 can also be used to predict $F_s$ maps at similar scales. However, compared to Kortum et al. (2024), our study focuses mainly on the prediction of meter-scale $F_s$ distributions (Sec. 3.3). In addition, we explored the effect of sea drift and deformation on the correlation between altimetric scans and SAR images. As shown in Section 3.2, third-party, large-scale drift products and local adjustments can be used to facilitate the collocation between the two. Related representation issues are further discussed in Section 4.3.

In Petty et al. (2017) the authors studied the statistical relationship between C-band backscatter measured by ASCAT and the variability of sea ice topography. The relationship is further used to estimate the atmospheric form drag coefficients based on backscatter maps. Although the scatterometers have relatively coarser resolution ($25km$ for ASCAT), the underlying mechanism of the topography-to-backscatter relationship is similar to our study. The macro-scale roughness of the sea ice cover (i.e., topography) and the sea ice type dependent surface properties affect microwave backscatter, resulting in the statistically significant relationship between the two.

### 4.3 Spatial and temporal locality of the statistical relationship between $F_s$ and $\sigma_0$

The statistical relationships between $F_s$ and $\sigma_0$ in Section 3.1.1 and 3.1.2 are based on OIB data and SAR images acquired on the same day. Furthermore, in Section 3.2, we demonstrated that there is large variability in this relationship, potentially caused by differences in sea ice/snow conditions and practical factors such as different observational geometries. Therefore, the statistical relationship is spatially localized, which implies that the extrapolation of freeboard measurements (e.g., Sec. 3.3) should be carried out locally.





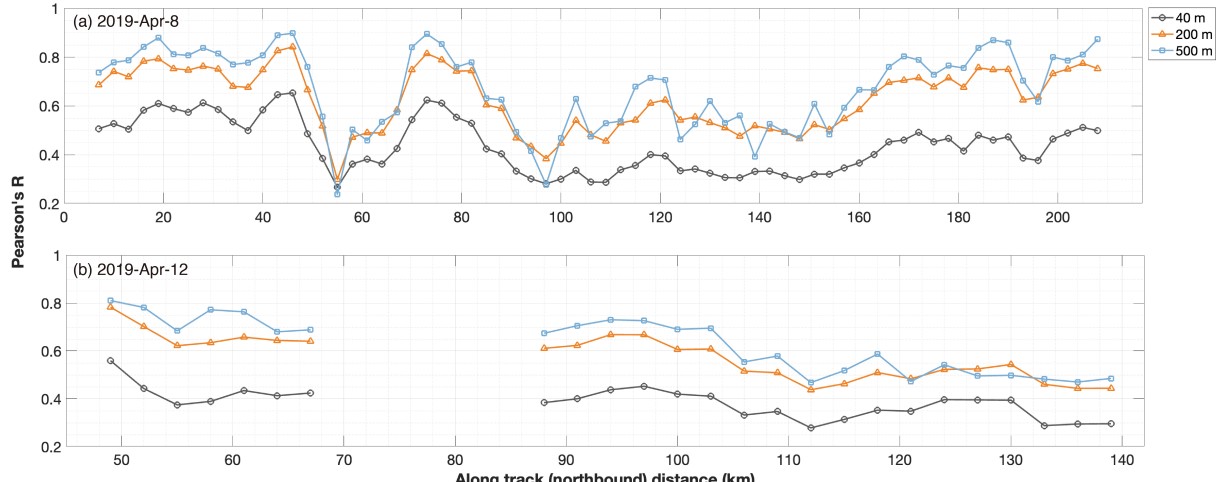

**Figure 11.** The statistical correlation between $F_s$ and $\sigma_0$ at three spatial scales: $40m$, $200m$, and $500m$. The coarsening is applied to both $F_s$ and $\sigma_0$ at these scales. The results for the OIB track on April 8th and 12th are shown in panel a and b, respectively. In order to accumulate enough samples, especially at the $500m$ scale, both the inbound and the outbound segments are used to compute the correlation coefficients. Note that in order to accommodate the effective resolution of $\sigma_0$ maps, in Fig. 2 and 3, we only applied spatial averaging to $F_s$ but not to $\sigma_0$.

Furthermore, we explore the temporal transferability of this relationship, by matching SAR images collected 1 week from the OIB sample segments. Correspondingly, sea ice may undergo significant drift and deformation, as well as thermodynamic changes during a week-long interval between the OIB and SAR observations.

350    For the sample segment on April 8th (Sec. 3.1.1), we use SAR images from April 1st and April 15th, and collocate both with the SAR image on April 8th and the $F_s$ map (Fig. S7). The analysis of the drift corrections indicates that there is negligible sea ice movement between April 8th and April 15th, and the statistical relationships between $F_s$ and $\sigma_0$ are consistent (Fig. S7, lower panels). However, the maximum correlation coefficient between $F_s$ and $\sigma_0$ is much lower at 0.4 for the SAR image on April 1st, as compared to 0.6 for April 8th (Fig. S7, upper panels). The drift corrections obtained from SAR images on April 1st and April 8th confirm significant sea ice deformation, leading to suboptimal collocation between not only SAR images, but also SAR and OIB (note the scattered samples in Fig. S7, panels b and c).

For the sample segment on April 12th (Sec. 3.1.2), SAR images from April 5th and April 19th are used for a similar analysis. Between April 5th and 12th, significant sea ice drift and deformation is present for the sea ice cover around the sample segment (Fig. S8.a). Correspondingly, the correlation coefficients between $F_s$ and $\sigma_0$ also witness significant drops: from 0.28 to 0.15

360    for the outbound segment, and from 0.54 to 0.45 for the inbound segment. On the contrary, between April 12th and 19th, sea ice drift is evident, but very small deformation is present, as indicated by the collocation of SAR images (Fig. S8.d). The correlation coefficients between $F_s$ on April 12th and $\sigma_0$ on April 19th largely remain the same as that based on April 12th. Specifically, the coefficient is 0.27 for the outbound segment and 0.54 for the inbound segment.




Both cases indicate that the collocation between OIB and SAR deteriorates at longer time intervals, and there are corre-
sponding drops in the statistical relationships. This is presumably caused by synoptic scale forcings that drive sea ice drift and
deformation, which reduce how well the SAR backscatter and OIB freeboard are co-located. As indicated by both observa-
tions and modeling studies (Marsan et al., 2004; Rampal et al., 2008; Ning et al., 2024), sea ice deformation is localized, and
multi-fractal both spatially and temporally. More importantly, there is strong coupling between the spatial and the temporal
domain. At longer time intervals, there is lower spatial localization of sea ice deformation, which potentially complicates the
collocating of SAR and altimetry scans. Furthermore, thermodynamic changes such as snowfall events, snow stratigraphic
changes, as well as newly formed sea ice ridges and leads, can also greatly modulate both $F_s$ and/or C-band backscatter(Tsai
et al., 2019; Manninen, 1992). These changes are also usually associated with synoptic events, which potentially co-occur
with sea ice drift and deformation. In summary, there is a strong locality in the statistical relationship between $F_s$ and $\sigma_0$. The
spatial and temporal windows for collocating SAR and altimetry scans and further upscaling the freeboard measurements is an
important research topic for future studies.

## 4.4 On the upscaling of IS2 measurements

Compared with the $1m$-scale $F_s$ maps from OIB, the standard sea ice elevation (ATL07) and freeboard (ATL10) products of IS2
are provided in beam profile segments. Since each beam segment consists of ~150 aggregated photons, the nominal resolution
is between 10 and $20m$ in the along-track direction for the three strong beams and ~11 m, the footprint diameter(Neumann
et al., 2020) in the across-track direction. For weak beams, the beam segment resolution is even coarser by approximately 4
times. By constraining and coarsening OIB $F_s$ maps to the footprints of IS2 strong and weak beam segments, we find that the
correlation maps between $F_s$ and S1 backscatter is in good agreement with those based on the full OIB segment (results for
the sample segments shown in Fig. S9). Therefore, the collocation with S1 images can also be carried out with IS2 elevation
measurements.

We re-apply the prediction algorithm in Section 3.3 to IS2 footprints of the sample segments. Specifically, the prediction
is trained and validated on the IS2 beam segments on the inbound and the the outbound OIB segments, which cover the IS2
beam pairs #1-#2 and #3-#4, respectively. However, compared to the $1m$-scale OIB $F_s$ map, the following limitations of IS2
are present: First, the IS2 beam segments are coarser, especially for the weak beams. Second, the IS2 ground coverage is much
narrower at $17m$, compared with the ~$1.4km$ width of the $F_s$ map. As a result, on the $9km$ sample segments, there is a very
limited number of IS2 beam segments (i.e., $\overline{F_s}$ samples). Therefore, in order to accumulate enough samples for prediction, we
extend the sample segments in both directions to $27km$ (equivalent to the length scale used in Fig. 6).

Specifically, we follow the three-step routine for the prediction and evaluation of $F_s$. First, by using IS2 beam segments
on the inbound segment (i.e., the #1-#2 beam pair), we bin the $F_s$ samples to $\sigma_0$, and further carry out the PDF fitting with
3-component Log-Logistic mixture model within each $\sigma_0$ bin. Second, we predict the $F_s$ distribution on the corresponding
outbound segment, using the $\sigma_0$ observations on the IS2 footprints (i.e., the #3-#4 beam pair). Finally, we validate the prediction
with the observed $F_s$ samples.





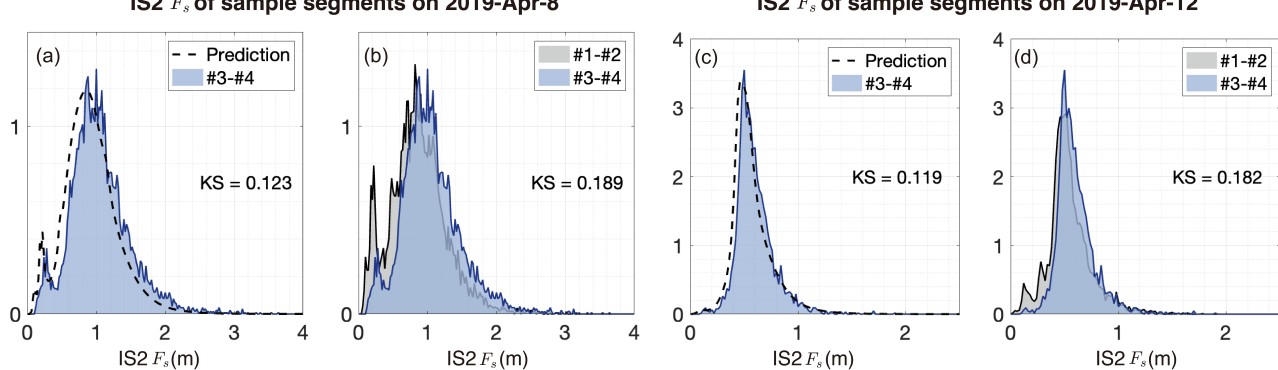

**Figure 12.** Same as Fig. 9, but for $F_s$ on IS2 beam segments on the sample segments on April 8th (panel a and b) and April 12th (panel c and d). Since there are limited number of IS2 beam segments, the length of the sample segments is enlarged to $27km$.

Figure 12 shows the results for the sample segments on April 8th and 12th. Similar to the validation of the $1m$-scale $F_s$ in Figure 9, the prediction on IS2 footprint also yields a good match with the observed $F_s$ distribution. In addition, the K-S distance is effectively reduced with the prediction: from 0.189 to 0.123 for the sample segments on April 8th, and from 0.182 to 0.119 for those on April 12th. Using the backscatter information over the prediction area produces an $F_s$ prediction that more closely matches the observed $F_s$ for Beams #3 and #4 than simply extrapolating the $F_s$ from Beams #1 and #2. Especially, the representation of thin ice (less than $30cm$ thick) has greatly improved for both cases, which is the major reason for the reduction in K-S distance.

## 5   Summary and Outlook

In this study we investigate the statistical relationship between sea ice freeboard and C-band microwave backscatter, by using collocated OIB observations and S1 images. Stronger SAR backscatter is observed for higher snow freeboard, which is attributed to the sensitivity of backscatter to both the sea ice type, with generally high volume scattering for MYI in winter, and ice topographic features such as ridges, with older ice having experienced stronger deformation (Krumpen et al., 2025). Moreover, the scale-dependency of this statistical relationship, along with its spatial and temporal locality, is further studied. A algorithm for predicting and extrapolating sea ice topographic measurements with SAR images is introduced that incorporates both: (1) the ICESat2 footprint size, and (2) the heteroskedasticity of sea ice total freeboard.

**Looking forward to basin-scale retrievals:** For the upscaling of IS2 observations at basin scale, concurrent and spatially collocated SAR images should be used, such as those from S1 and the RadarSat Constellation Mission (RCM, see: MDA, 2021). Specifically, we have demonstrated both spatial and temporal locality of the derived statistical relationships. For altimetry and SAR observations that are separated by long temporal intervals, thermodynamic and dynamic processes within the ice and overlying snow can degrade the relationships between macro-scale topography and C-band backscatter. Another key factor is the spatial scale for the upscaling of IS2 measurements. In Section 3.3 the prediction is designed to incorporate meter-



scale $F_s$ maps. The photon-based elevations represent a similarly fine spatial scale to the OIB ATM, but contain considerable
uncertainties. Also, the different photon rates over various sea ice surfaces should also be accounted for. The proper temporal
and spatial scales for the matching SAR images and the upscaling of IS2 measurements should be the subject of detailed studies
in the future.

**Historical & future campaigns:** The sea ice topographic roughness and the statistical fittings are dependent on the scale of
altimetric observations (Sec. 3). Beyond the OIB ATM scans ($1m$-scale) and the IS2 beam segments ($20 \sim 30m$ for the strong
beams), various historical and future campaigns feature drastically different payload design and resolutions. For example,
the nominal footprint size of ICESat is $65m$ (Farrell et al., 2009), and at this scale there also exist statistically significant
relationships between $F_s$ and the C-band backscatter(Kortum et al., 2024; Macdonald et al., 2024). Besides, the concurrent SAR
observations at both C- and L-bands, such as ALOS(Advanced Land Observing Satellite) and ALOS-2(Shimada et al., 2009;
Kankaku et al., 2013), can be further used for the study of the relationships and potentially upscale altimeter measurements.
For ICESat, by combining with data from SAR satellite payloads such as ESA's EnviSat ASAR (Miranda et al., 2013), the
upscaling of ICESat can be carried out for constructing a wider coverage record of sea ice freeboard for the period 2003–2008.

*Data availability.* The data from OIB campaigns in April, 2019 are available from the National Snow and Ice Data Center: https://nsidc.org/
data/ilatm1b/versions/2, and https://nsidc.org/data/ilnsa1b/versions/2 (last access: 6 September 2024). S1 EW images are accessed from the
Copernicus Data Space Ecosystem (available at https://browser.dataspace.copernicus.eu/, last access: 6 September 2024) and processed them
using the ESA Sentinel Application Platform (SNAP) toolbox. The complete list of used SAR images are provided in the supplement with
public access. The ATL07 and ATL10 product from ICESat-2 (version 6) are accessed at the National Snow and Ice Data Center through
https://nsidc.org/data/atl07/versions/6 and https://nsidc.org/data/atl10/versions/6 (last access: 6 September 2024). The OSI-SAF sea ice drift
product is available at: https://osi-saf.eumetsat.int/products/osi-455 (last access: 6 September 2024). DTU15MSS_1min can be found at:
https://www.space.dtu.dk/ (last access: 12 February 2025).

The interpolated and stitched $1m$-resolution total freeboard fields (in $3km$ segments) of the sample segments on 2019-Apr-8 and 2019-
Apr-12 are achieved at: https://zenodo.org/records/14930672 (last access: 26 February 2025). Additionally, the sea ice type maps based on
Sentinel-1 EW images can also be accessed at the same URL.

## Appendix A: Processing of OIB ATM elevations

The elevations of the original ATM samples are converted into the total freeboard (or the snow freeboard, denoted $F_s$). For OIB
flights on April 8th and 12th which were organized into racetracks (Fig. 1), we merge all OIB samples to construct a merged
map of $F_s$ for both the northbound and the southbound flight passes. Specifically, two steps are carried out, as follows.

## A1   Construction of the per-pass $1m$-scale $F_s$ map

As the first step, for each OIB pass, we converted OIB ATM samples into the $F_s$ map which covers over $500m$ across the
OIB flight path. Both wide scan and the narrow scan of the OIB ATM are utilized. For a local segment along the OIB flight





(e.g., $10km$ in length), we first project each ATM sample under the polar stereographic projection according to its geolocation (i.e., its latitude and longitude). Then, we interpolate the samples into a $1m$-scale elevation map, using linear interpolation. Afterwards, we apply atmospheric and tidal corrections to the elevation based on mean sea-surface height (DTU15 MSS model). Finally, we treat the corrected elevation as elevation anomalies, and apply the lowest elevation method to retrieve the freeboard. Specifically, the lowest 1‰ of elevation samples within each $10km$ segment are extracted and linearly interpolated to construct the local water level (also at $1m$-scale) using the Inverse Distance Weighting (IDW) method. The final $1m$-scale

$F_s$ map is further validated with the standard $40m$-scale $F_s$ product from IDCSI (Fig. S1).

## A2  Collocation between OIB passes and the construction of the merged $F_s$ field

We further merge the three OIB passes to form the $F_s$ map that covers over $1.4km$ across the flight path. Since the central pass and the left pass were separated by 1∼2 hours, and the central pass and the right pass by 3∼4 hours, the sea ice cover potentially had undergone drift and deformation. Therefore, we first search for corrections between each of the two pairs of

OIB passes. For each $3km$ segment, we maximize the correlation of the overlapping part of the $F_s$ maps of the central and the left (or the right) pass, by adjusting the relative location of the left (or the right) pass with respect to the central pass. After the maximum correlation is attained, we record the corrections in both the along-track and the cross-track directions, and further merge the left and the right pass to the central pass, in order to form a unified $F_s$ map. In Figure 2.a (3.a) we show the merged $F_s$ maps for the sample segment on April 8th (12th), and in Figure S2 (S4) the correlation maps between OIB passes.

For certain segments, the central pass and the left (or right) pass do not overlap, and therefore they are not included in further analysis (especially in Fig. 5). Figure A1 and A2 show the corrections and the maximized correlation of $F_s$ maps between OIB passes for all $3km$ segments on April 8th and 12th, respectively. For April 8th, very high correlation coefficients were attained for all segments (Pearson's $r$ all over 0.94). Besides, meter-scale corrections were required, which potentially arise from locating uncertainties. On the contrary, on April 12th, evident corrections with length over $100m$ were needed to maximize

the correlation, which are also consistent with the large-scale drift provided by OSI-SAF (details not shown). Therefore, we consider these corrections are associated with sea ice drifts. Evident changes of the sea ice drift at the location of $120km$ along the OIB flight path is detected for both the inbound and the outbound flights, indicting the presence of sea ice deformation. Especially, the correlation coefficients for the $3km$ segments also dropped to lower than 0.9 where the deformation is detected. Collocation and the resulting correlation coefficients at the scale of $500m$ around the location of of the deformation further

indicate that the deformation are localized (i.e., within $500m$) and present at several along-track locations (Fig. A2).

## Appendix B: S1 EW images used for analysis for OIB campaigns

The sea ice classification algorithm used in this study is based on: Lohse et al. (2020, 2021); Guo et al. (2023). Lohse et al. (2020) developed a supervised algorithm that accounts for the class-dependent IA effects, known as the GIA classifier. While this classifier performs well in addressing IA sensitivity, some misclassifications and ambiguities remain. To address these

**Figure A1.** Collocation between different OIB flight passes on April 8th, 2019. The along-track segment length is $3km$. The local corrections of the left and the right pass with respect to the middle pass for each segment on the outbound (inbound) flights is shown in panel a and b (g and h), respectively. The correlation coefficients (Pearson's $r$) after the collocation between the left and the middle pass and that between the right and the middle are shown in panel c and d the for the outbound flight, respectively. Similarly, panel e and f show the results for the inbound flights.





**Figure A2.** Same as Fig. A1, but for the OIB campaign on April 12th, 2019. Correlation coefficients lower than 0.8 are marked by filled symbols in panel c, d, g and h. For segments around the apparent deformation (at $\sim 120 km$ along the track), the local drift correction is further refined to $500m$ in the along-track direction. The $500m$-scale drift corrections and the correlation coefficients are marked by circles and thin lines.



issues, Lohse et al. (2021) and Guo et al. (2023) enhanced the algorithm by incorporating GLCM texture features, resulting in improved class separation. This study uses this classification approach to produce sea ice type maps on the selected S1 scenes.

In the classification process, seven GLCM textures are derived from the HH channel of each SAR image, with a texture window size of 11 pixels. Then, SAR intensities (HH and HV) and GLCM textures (HH) are used as input to the GIA classifier, which incorporates their IA dependencies. Sea ice is classified into three types: level first-year ice (LFYI), deformed first-year

ice (DFYI), and multiyear ice (MYI). To further refine the results, a Markov Random Field based contextual smoothing process is applied with a window size of 3 pixels (Doulgeris, 2015). The final sea ice type maps have a pixel size of $40\,m$, but their effective spatial resolution is significantly coarser due to SAR speckle filtering and textural processing.

Table B1 lists all the S1 EW images used in this study, specifically collected during the OIB campaigns on April 8th and 12th. Two types of images are included: those on the adjacent days of the campaigns, and those separated by about 1 week

from the campaigns. The corresponding IS2 reference ground tracks (RGT) are also shown.

*Author contributions.* SX carried out conceptualization of the study. SL processed the OIB dataset. WG processed S1 images and provided sea ice type maps. SL, WG, YF, SX carried out the analysis, with input from other authors. All author contributed to the writing of the manuscript.

*Competing interests.* The authors declare that they have no competing interests.

*Acknowledgements.* This work is mainly supported by the joint project of INTERAAC, co-funded by the National Key R&D Program of China (grant no.: 2022YFE0106700) and the Research Council of Norway (grant no.: 328957). JCL is partially supported by the SUDARCO (Forskning for god forvaltning av Polhavet) project under the Fram Centre (#2551323), the DynAMIC (Detecting episodes of Arctic sea ice Mass Imbalance) project under RCN (#343069), and the SI/3D (Summer Sea Ice in 3D) project under the European Research Council, ERC (#101077496). SX is also partially supported by the National Natural Science Foundation of China (grant no.: 42030602) and the

International Partnership Program of Chinese Academy of Sciences (grant no.: 183311KYSB20200015).



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




**Table B1.** OIB campaign and the corresponding S1 images. The corresponding ICESat2 ground tracks' information, including its visit times are shown in the last column.

| OIB ATM data | S1 image | IS2 RGT |
|---|---|---|
| | 2019-Apr-07: S1B_EW_GRDM_1SDH_20190407T150052_20190407T150152_015702_01D768_1E98 | |
| | 2019-Apr-07: S1B_EW_GRDM_1SDH_20190407T145952_20190407T150052_015702_01D768_0AEC | 2019-Apr-08: |
| 2019-Apr-08: | 2019-Apr-08: S1B_EW_GRDM_1SDH_20190408T140254_20190408T140354_015716_01D7D4_334A | RGT 0157 |
| 12:24:18 to 15:51:59 | 2019-Apr-09: S1B_EW_GRDM_1SDH_20190409T144345_20190409T144445_015731_01D856_468A | Beam 1,2,3,4 |
| | 2019-Apr-01: S1B_EW_GRDM_1SDH_20190401T141105_20190401T141205_015614_01D465_4CC6 | 13:09:59 to 13:10:39 |
| | 2019-Apr-15: S1A_EW_GRDM_1SDH_20190415T144457_20190415T144602_026802_030317_2C1F | |
| | 2019-Apr-12: S1B_EW_GRDM_1SDH_20190412T182436_20190412T182536_015777_01D9D0_7AB9 | |
| | 2019-Apr-11: S1B_EW_GRDM_1SDH_20190411T174333_20190411T174433_015762_01D955_0683 | 2019-Apr-12: |
| 2019-Apr-12: | 2019-Apr-13: S1B_EW_GRDM_1SDH_20190413T190536_20190413T190636_015792_01DA51_7539 | RGT 0218 |
| 13:11:18 to 15:49:17 | 2019-Apr-05: S1B_EW_GRDM_1SDH_20190405T201050_20190405T201154_015676_01D68A_61C3 | Beam 1,2,3,4 |
| | 2019-Apr-19: S1B_EW_GRDM_1SDH_20190419T195430_20190419T195534_015880_01DD4B_40E2 | 13:03:21 to 13:03:54 |