# Peer review of "On the Statistical Relationship between Sea Ice Freeboard and C-Band Microwave Backscatter – A Case Study with Sentinel-1 and Operation IceBridge"

_EGUsphere, 2025_

## Referee Comment (RC1)

**On the Statistical Relationship between Sea Ice Freeboard and C-Band Microwave Backscatter – A Study with Sentinel-1 and Operation IceBridge**

The authors present a study on the local extrapolation of freeboard statistics using Sentinel-1 backscatter. I found the paper to be well written and researched. The results add value and new insight into a recently emerging topic in the scientific discourse concerning the combination of altimetry and SAR instruments. The methodologies used are well suited to the investigation.

Overall, I think the paper certainly merits publication and should be ready for it after a few additions/clarifications.

Major:

1. **Concerning smaller scales:**
   I think it would be beneficial for the evaluation of the extrapolation scheme to also assess the accuracy of the predicted freeboard distribution on a smaller scale or per-pixel basis. Especially concerning the ICESat-2 results, the 27km segment length is quite large and it would be very interesting to know how well the relationships from HH brightness to freeboard distribution hold on a smaller scale. It would also make it easier to compare the accuracy of this extrapolation to existing methods (Macdonald 2024, Kortum 2024).

Minor:

2. **Concerning the matching of OIB/ICESat-2 and Sentinel-1:**

   a. I think the time differences between the different acquisitions should be mentioned, especially when discussing the coregistration/collocation. Furthermore, I think it would be beneficial to plot the corrections (meters offset) along the ICESat-2 track. Also, the size of segments used for the matching adjustment should be mentioned (is it the same 3 km as used for the inter OIB matching found in the Appendix?).

   b. I am quite surprised how good the linear correlations are (after correction) between OIB freeboard and sigma_0(HH). These are in large part significantly higher than reported in previous studies (Cafarella 2019, Segal 2020, Macdonald 2024, Kortum 2024). Maybe you could also give the regression without binning for easier comparison with these existing studies.

3. **HV vs HH:**
   In the supplement I see that the observed HV correlations seem similar/stronger than HH, yet in the manuscript you focus entirely on the HH band. I suppose you deem the HH band to be more robust, due to the better signal to noise ratios – but this in my opinion should be reasoned before continuing only with the HH band.

4. **Paper summary:**
   In the Kortum et al 2024 paper, extrapolation is done not for temporally coincident scenes,

but with an allowance of up to 24 hours in time difference between Sentinel-1 and ICESat-2 measurements.

5. **Segment sizes:**
   As scale is quite important with these observations, I think the dimensions of segments (and size of bins) should be mentioned in every figure where they appear, as it is sometimes not obvious at the moment.

6. **Linear regression:**
   A large part of the paper deals with linear relationships between backscatter and freeboard. It should be briefly reasoned somewhere when or why this is a reasonable assumption: It seems the relationship becomes more linear at larger scales for example, but at smaller scales linearity might be stronger simplification.

**Spelling etc.:**

There should be a small non-breaking space between units and numbers (10 m not 10m).

L.62 'organised into racetracks' – I am not familiar with this expression. Maybe 'organised in a racetrack pattern' is easier to understand.

L.183 The segment size (I believe 9 km) should be mentioned here.

Fig. 4: I believe it should be 'Binning' not 'Bining'

L.331: 'sea **ice** drift'

---

## Author Comment (AC1)

**Reply to Dr. Karl Kortum (Referee #1)'s comments (RC1)**

The authors would like to thank Dr. Karl Kortum for the invaluable comments and suggestions. The following are the replies for each point of the comment, together with specific revisions that are made. The original comments are in *blue italic* font and listed in paragraphs, with our reply following each paragraph separately. The revisions are also highlighted in the revised manuscript in blue and marked by ***RC1***.

*The authors present a study on the local extrapolation of freeboard statistics using Sentinel-1 backscatter. I found the paper to be well written and researched. The results add value and new insight into a recently emerging topic in the scientific discourse concerning the combination of altimetry and SAR instruments. The methodologies used are well suited to the investigation.*

*Overall, I think the paper certainly merits publication and should be ready for it after a few additions/clarifications.*

*Major:*

*1. Concerning smaller scales:*
*I think it would be beneficial for the evaluation of the extrapolation scheme to also assess the accuracy of the predicted freeboard distribution on a smaller scale or per-pixel basis. Especially concerning the ICESat-2 results, the 27km segment length is quite large and it would be very interesting to know how well the relationships from HH brightness to freeboard distribution hold on a smaller scale. It would also make it easier to compare the accuracy of this extrapolation to existing methods (Macdonald 2024, Kortum 2024).*

**Reply**: Thanks for the helpful comments. We would like to first make clarifications on the predictions we carry out. Specifically, we have developed two prediction algorithms. The first prediction method is the traditional, "pixel-wise" prediction of the freeboard, similar to Macdonald et al. (2024) and Kortum et al. (2024). This method yields freeboard maps, and it is introduced in Sections 3.1 and 3.2.

The second method targets at the prediction of the freeboard distribution at the altimeter's native resolution (e.g., 1-meter for OIB ATM, covered in Sec. 3.3). For instance, with OIB ATM the freeboard distribution's resolution is 1m, while for IS2 (ATL07) the resolution is the beam segments' nominal size (i.e., about 30 m for the strong beams). The second method is based on: (1) binning freeboard samples according to backscatter, and (2) the fitting of freeboard distribution in each bin. Therefore, this method is free from regression models and their potential problems.

For either method, we accumulate collocated samples between the altimeter and the SAR image to train the prediction model. And the samples are from a local region of 9km (along-track). Hence 9km is not the resolution of the freeboard map (for the first method).

For the traditional method which predicts the freeboard map, in the manuscript we did not show the prediction results, but we did show the regression model's parameters (Fig. 6).

Practical resolution of freeboard maps with Sentinel-1 images is about 200m, at which the regression model's R2 is sufficiently high. As pointed out by the referee, this method is comparable to existing ones in Macdonald et al. (2024) and Kortum et al. (2024).

Therefore, according to the referee's suggestion, we have conducted a comparative analysis of the statistical relationships across these studies. It should be noted that the sampling scale in existing literature is based on basin-scale observations, which is much larger than ours (at 9km or 27km). To improve comparability, in this comparison, we extended the sampling scale for our training data to the entire OIB track (~200 km in length). We further aligned the resolution of the statistical relationships with the other two works. The comparison results are presented in the following table:

| R | 40m Fs VS 40m backscatter | | | 100m Fs VS 100m backscatter | | |
|---|---|---|---|---|---|---|
| | *Macdonald et al. 2024* | This study | | *Kortum et al. 2024* | This study | |
| | | April 8 | April 12 | | April 8 | April 12 |
| HH | \ | 0.48 | 0.40 | 0.49 | 0.58 | 0.47 |
| HH(FYI) | 0.37 | 0.53 | 0.44 | 0.18 | 0.63 | 0.50 |
| HH(MYI) | 0.48 | 0.40 | 0.24 | 0.34 | 0.49 | 0.30 |
| HV | \ | 0.49 | 0.47 | 0.62 | 0.58 | 0.59 |
| HV(FYI) | 0.40 | 0.53 | 0.50 | 0.32 | 0.62 | 0.60 |
| HV(MYI) | 0.49 | 0.41 | 0.37 | 0.49 | 0.50 | 0.48 |

There are two spatial scales for comparing our results with existing studies: 40m (for comparing with Macdonald et al. (2024)) and 100m (for comparing with Kortum et al. (2024)). In both comparisons, we found that the correlation of the freeboard with the HV-channel backscatter is slightly higher than with the HH-channel. The correlations (Pearson's *r*) in our study are broadly consistent with those reported in Macdonald et al. (2024), ranging from 0.37 to 0.53. This consistency provides confidence in the reliability of our results. And we also observe that the correlation for MYI using HH backscatter on April 12, 2019 is lower than correlation in Macdonald et al. (2024). Our correlations are higher than those presented in Kortum et al. (2024). The discrepancy may be due to the differences in study area, sampling scale, and SAR data preprocessing workflows, among other factors.

*Minor:*

*2. Concerning the matching of OIB/ICESat-2 and Sentinel-1:*
*a. I think the time differences between the different acquisitions should be mentioned, especially when discussing the coregistration/collocation. Furthermore, I think it would be beneficial to plot the corrections (meters offset) along the ICESat-2 track. Also, the size of segments used for the matching adjustment should be mentioned (is it the same 3 km as used for the inter OIB matching found in the Appendix?).*

**Reply**: First, we agree with your suggestion that it is necessary to mention the time differences between the different acquisitions in the manuscript. As Table B1 presents the

observation times of OIB, ICESat-2, and Sentinel-1, the acquisition time differences are shown below:

- OIB passes collocation:
  - 1.25 hours between left and middle pass;
  - 2.5 hours between right and middle pass.
- OIB-Sentinel-1 collocation:
  - 2019-04-08 track: ~40 minute separation;
  - 2019-04-12 track: ~4 hour separation.

We had added these details in the revised manuscript.

Second, the drift corrections (in meters) along/across the ICESat-2 track are displayed in the correlation map (Fig. S9, panels a, b, e, and f).

Third, regarding the size of segments used for collocation, the 3km segment length is used for OIB pass collocation in Appendix B2, while the 9 km segment length is used for the collocation between OIB and Sentinel-1. We have further cleared this information to the revised manuscript and corresponding figures.

*b. I am quite surprised how good the linear correlations are (after correction) between OIB freeboard and sigma_0(HH). These are in large part significantly higher than reported in previous studies (Cafarella 2019, Segal 2020, Macdonald 2024, Kortum 2024). Maybe you could also give the regression without binning for easier comparison with these existing studies.*

**Reply**: In response to your comment, we would like to highlight that Figure 6 provides all the the linear regression parameters and the corresponding Pearson's *r* (**before binning**) between the 200m-scale freeboard and 40m-scale $\sigma 0$ for all segments on April 8th (panels a, b, and c) and April 12th (panels d, e, and f).

*3. HV vs HH:*
*In the supplement I see that the observed HV correlations seem similar/stronger than HH, yet in the manuscript you focus entirely on the HH band. I suppose you deem the HH band to be more robust, due to the better signal to noise ratios – but this in my opinion should be reasoned before continuing only with the HH band.*

**Reply**: Thanks for the helpful comment. Indeed, we have also analyzed the statistical relationship between freeboard and backscatter in the HV-channel for the whole OIB track. Our results show general consistency with previous studies (Macdonald et al., 2024; Kortum et al., 2024), that freeboard generally correlates slightly better with the HV-channel than with the HH-channel backscatter. Besides, HV-channel in C-band has been shown to be crucial for the FYI-MYI classification (Komanov & Buehner, 2019), which can be combined with HH-channel for future studies of their relationship to ice topography.

The main reasons we focus on the HH-channel in this study are:
1. The HV-channel backscatter is generally much weaker than the HH-channel. This is particularly evident for FYI, where HV backscatter often falls below the nominal

noise floor (Segal et al., 2020). Additionally, the sub-swath artifacts are more evident (i.e., on sub-swath boundaries) for the HV-channel for Sentinel-1 EW mode images (Lohse et al., 2021).

2. Despite the stronger correlation observed in the HV band, the qualitative statistical relationship between freeboard and backscatter is similar when using either the HH or HV channel.

Given these considerations, we have conservatively focused on the S-1 HH channel in our manuscript.

Segal RA, Scharien RK, Cafarella S, Tedstone A. Characterizing winter landfast sea-ice surface roughness in the Canadian Arctic Archipelago using Sentinel-1 synthetic aperture radar and the Multi-angle Imaging SpectroRadiometer. Annals of Glaciology, 61(83):284-298. doi:10.1017/aog.2020.48, 2020.

Lohse, J., Doulgeris, A. P., and Dierking, W.: Incident Angle Dependence of Sentinel-1 Texture Features for Sea Ice Classification, Remote Sensing, 13, https://doi.org/10.3390/rs13040552, 2021.

Komarov A. S. and M. Buehner: Detection of First-Year and Multi-Year Sea Ice from Dual-Polarization SAR Images Under Cold Conditions. IEEE Transactions on Geoscience and Remote Sensing, 57(11), 9109-9123, doi: 10.1109/TGRS.2019.2924868, 2019.

*4. Paper summary:*
*In the Kortum et al 2024 paper, extrapolation is done not for temporally coincident scenes, but with an allowance of up to 24 hours in time difference between Sentinel-1 and ICESat-2 measurements.*

**Reply**: It has been revised in the manuscript.

*5. Segment sizes:*
*As scale is quite important with these observations, I think the dimensions of segments (and size of bins) should be mentioned in every figure where they appear, as it is sometimes not obvious at the moment.*

**Reply**: Thanks, we have revised the figures accordingly.

*6. Linear regression:*
*A large part of the paper deals with linear relationships between backscatter and freeboard. It should be briefly reasoned somewhere when or why this is a reasonable assumption: It seems the relationship becomes more linear at larger scales for example, but at smaller scales linearity might be stronger simplification.*

**Reply**: We agree that there are limitations of using the linear regression model. However, there are several reasons, listed below:

1. The linear regression model is simple and reduces the potential of overfitting. We recognize that complex microwave backscattering mechanisms drive the differences in the SAR-measured backscatter among the various sea ice types (FYI, MYI, level and ridged ice, etc). Hence the relationship between freeboard and backscatter should not be linear. However, we don't have a comprehensive understanding of these mechanisms. Consequently, we adopted a linear fit as a first-order approximation to this relationship.

2. Our analysis reveals that the relationship between freeboard and backscatter becomes much more linear at larger spatial scales (e.g., 200m-scale as in Fig. 2 and 3). Moreover, it is remarkably linear after binning to the backscatter (panel c and e in Fig. 4 and 5). This ensures that the linear regression model is highly effective in predicting the mean freeboard.

3. Lastly, and arguably more importantly, in this study we focus on predicting the freeboard distribution, rather than the pixel-scale freeboard map. It is worthy to differentiate these two types of prediction tasks. The regression model is needed for the prediction of the freeboard map (i.e., point to point prediction). However, it is not needed for the prediction of freeboard distribution in Sec. 3.3. The prediction of the freeboard distribution allows us to fully utilize the altimeter's measurements at their native resolution (e.g., meter-scale of the OIB ATM). Therefore, given our focus of the study, we only adopt the linear regression model in the analysis of Sec. 3.1 and 3.2.

*Spelling etc.:*
*There should be a small non-breaking space between units and numbers (10 m not 10m).*

**Reply**: It has been revised in the manuscript.

*L.62 'organised into racetracks'*
*– I am not familiar with this expression. Maybe 'organised in a racetrack pattern' is easier to understand.*

**Reply**: Agree. This has been revised in the manuscript.

*L.183 The segment size (I believe 9 km) should be mentioned here.*

**Reply**: It has been revised in the manuscript.

*Fig. 4: I believe it should be 'Binning' not 'Bining'*

**Reply**: Thanks, we have revised the figures accordingly.

*L.331: 'sea ice drift'*

**Reply**: It has been revised in the manuscript.

---

## Author Comment (AC2)

**Reply to Referee Comment #2 (RC2)**

The authors would like to thank the referee for the invaluable comments and suggestions. The following are the replies for each point of the comment, together with specific revisions that are made. The original comments are in *green italic* font and listed in paragraphs, with our reply following each paragraph separately. The revisions are also highlighted in the revised manuscript in green and marked by **RC2**.

*Dear TC editor and authors of the manuscript egusphere-2025-1069,*

*The manuscript topic is interesting for sea ice monitoring applications.*
*In its current form I can not recommend publication of this manuscript.*
*There exist significant deficiencies in the manuscript.*
*Therefore, I recommend a major revision before considering publication.*

*The major deficiencies that need to be addressed are:*

*A small amount of data: only 11 SAR images and these data along the measurement lines have been used in the study. It would be good to have more data included to be able to provide more general results. In its current for the study is both locally and temporally very restricted and remains only a case study.*

*If it is not possible to involve more data, it must be emphasized that this is a case study, also in the title, instead of "study" use "case study".*

**Reply**: As suggested, we have revised the tite as a "case study" of the relationship between sea ice topography and microwave backscatter. The revised manuscript has the title: "*On the Statistical Relationship between Sea Ice Freeboard and C-Band Microwave Backscatter - A Study with Sentinel-1 and Operation IceBridge*"

*Can any general conclusions be made based on this analysis? Future work should concentrate on analyzing larger data sets with different weather and ice conditions.*

**Reply**: We thank the reviewer for the comment, and we believe that the analysis with OIB data and SAR images does indicate that there exist statistical relations between ice topography and C-band microwave backscatter. We totally agree that analysis with larger datasets is necessary and it is planned for future work.

Besides, we also want to emphasize the major points of our study:
1. Our analysis covers a range of ice regimes in the Arctic, including thick and thin multi-year as well as thin first-year ice. For all these types, there exists a statistically significant relationship between the backscatter and freeboard. We believe this is a new and important advance for our field.
2. The reprocessing of OIB ATM data allows us to examine this relationship at various spatial scales, from meter-scale freeboard distribution, to typical SAR pixel sizes, and

reaching scatterometer-relevant, kilometer-scales. This serves as the basis for working with different altimeter and SAR payload combinations.

3. We examine the locality of the statistical relationship, which is shown to be affected by sea ice type mixture spatially, as well as sea ice deformations due to large temporal separation. This information is key for large-scale synergy between altimeter and SAR observations.

4. One major focus for the freeboard prediction is on the freeboard distribution of the altimeter's native resolution. For OIB-ATM, the prediction is on the meter-scale, and for the prototype study with IS2, the prediction is on the scale of beam segments (~30m).

*The need for future work to extend the study and directions of the future work taking thes aspects into account should be clearly mentioned in the concluding section (summary and outlook).*

**Reply**: Based on the referee's suggestion, in the revised manuscript in the conclusion section, we will highlight the following key directions for future research:

1. Future work will incorporate a larger dataset to investigate the freeboard-backscatter relationship for regions with more first-year ice across seasons/stage of development (FYI). In this study, we have explored this relationship for MYI-dominated regions. So we plan to extend the study with the historical OIB campaigns with Sentinel-1 collocation, which covered various sea ice types and different areas of the Arctic basin.

2. We will also examine how the statistical relationship between freeboard and backscatter varies under different weather conditions. Factors such as melt conditions and heavy snowfall could potentially alter both the backscatter and the overall snow budgets. We will further study the changes and potential degradation of the freeboard-backscatter relationship.

*Weather condition data, except for the wind data has not been included. SAR backscatter is significantly dependent on the ice or snow surface temperature and they are naturally dependent on the air temperature history before the data acquisition. Therefore, I propose to include this information and emphasize on what kind of ice and snow (on ice) conditions the proposed results are useful.*

**Reply**: we have examined the weather conditions around April 8th and 12th when OIB campaigns were carried out (from April 1st to 16th). These weather conditions are based on the atmospheric reanalysis of ERA5. Hereby we summarize the weather conditions as follows:

1. For both campaigns, the atmospheric conditions were cold (Tair < 16degC), typical of the condition in April in the surveyed regions.
2. The SLP fields around the region surveyed on 8th indicate very weak wind forcing on the sea ice, with limited sea ice drift. For the surveyed region on 12th, the (geostrophic) wind is weak during the days around 12th, with no evident sea ice drift.

3. The precipitation during the whole week in the OIB-surveyed regional before the campaign is minimal (SWE<3mm) for both campaigns.

From these results, we draw the conclusion that the large-scale atmospheric conditions are typical of the later-winter condition in the respective regions. There were no sudden warming events and evident precipitation that potentially changes the SAR backscatter signature of the sea ice.

Moreover, at C-band, the SAR backscatter is dominated by the wavelength-scale roughness at the bottom of the snow cover. However, at X-band which the wavelength is much shorter, the backscatter is much more sensitive to snowfall and stratigraphy changes such as warming and refreezing.

**Air temperature (daily mean, degC):**

[Figure]

**Sea level pressure (daily mean, filled contour, in Pa) and large-scale sea ice drift (arrows, from OSI-SAF LR product, 48hr):**

[Figure]

**Total precipitation (weekly accumulated SWE, in mm)**

[Figure]

*It seems that the results for the HV channel are presented in the supplement. Why? The HV results should be part of the manuscript. Also, the information in appendix A and B could be included in the manuscript sections and leaving the appendices out.*

**Reply**: We thank the referee for pointing out the importance of the analyses with the HV channel. The motivation of providing them in the supplementary instead of the main manuscript is its conciseness.

We have incorporated the HV channel results into a newly added appendix, including Figure S3 and Figure S5 (which were in the supplementary file), along with the HV relationship corresponding to Fig. 6.

Regarding Appendix A, we understand the importance of clarity and conciseness in the manuscript. Appendix A focuses on the collocation between OIB passes, which is a critical part of our data preprocessing. While Figures A1 and A2 are large and primarily serve to support the correct collocation between OIB passes, their direct contribution to the main narrative of the manuscript is limited. Therefore, to prevent the manuscript from becoming lengthy, we have decided to retain Appendix A as appendix material.

*The methods should be persented in detail in a specific section (named e.g. "Methods" or "Methodology"). This should be after Section 2, i.e. Section 3.*

**Reply**: According to the suggestion, we add a new section titled "*Methodology*", which is Section 3 of the revised manuscript. This section provides a detailed description of the

methods for the statistical analyses of the relationship between freeboard and backscatter, as well as the method for predicting freeboard distribution using σ0 maps.

*It is not very clear how the SIT distribution prediction based on SAR sigma0 is exactly performed. Please, describe this essential phase in detail. All methods/algorithms should be described in detail in a specific methodology section.*

**Reply**: The detailed description of the method for predicting freeboard distribution is included in the newly added "*Methodology*" section.

*More detailed comments:*

*How the data is divided into independent training and test data sets for the regression.*

*This should be described in detail.*

**Reply**: The OIB flights were organized into racetracks, with the outbound (north-bound) and the inbound (south-bound) paths covering adjacent, but different sea ice cover. We use the inbound flight data (OIB-ATM and collocated SAR) for the training, and the outbound flight data for testing the regression. Therefore, the training and the testing use independent data. This information is now described in detail in the "Methodology" section of the revised manuscript.

*Incidence angle is mentioned and in the ice type classification it is taken into account.*

*However, it seems IA has not been taken into account in the sigma0 analysis. Would there be any effect if e.g. a simple linear incidence angle correction were applied (different slopes for sea ice C-band HH and HV sigma0 can be found in literature)?*

**Reply**: We agree that with a simple incident angle correction, the regression relationship will quantitatively change. Given that the segment length of 9km is used for our analysis, the correction induced change in Sigma0 within the segment is considered small. Therefore, the correction will change the intercept of the regression model, but not its slope. However, if a more complex angle correction scheme (such as the sea ice type dependent correction) is applied, both the slope and the intercept of the regression model are susceptible to changes.

In this study we refrain from using an incident angle correction mainly for two reasons:

(1) The incidence angle dependence of Sigma0 is clearly type dependent, which is different between FYI and MYI, as well as between level ice and ridged ice. A simple correction is not sufficient to account for these factors (see also Lohse et al. (2020), Kwok et al. (1992), etc.).
(2) The *optimal* incidence angle for the correction is unknown for deriving the statistical relationship between Sigma0 and freeboard. Whether it should be the near range

(i.e., 20deg) or the far range (i.e., about 50deg) in the SAR image's swath (Sentinel-1 EW mode) should be studied further in future work.

We do notice that the incidence angle dependency is relatively weaker for HV backscatter (Lohse et al., 2020). As a result, the changes in the regression parameters (i.e., the slope and the intercept) after incidence angle correction at HV channel is expected to be smaller than those at HH channel.

*Lohse J, Doulgeris AP, Dierking W. Mapping sea-ice types from Sentinel-1 considering the surface-type dependent effect of incidence angle. Annals of Glaciology. 2020;61(83):260-270. doi:10.1017/aog.2020.45*

*Ronald Kwok, Eric Rignot and Benjamin Holt. Identification of Sea Ice Types in Spaceborne Synthetic Aperture Radar Data, Journal of Geophysical Research, 97(C2), 2391-2402, 1992*

*At HV band it is well-known that S-1 has a significant noise pattern in range direction, especially near the boundaries of the subswaths. Also, some scalloping noise in the azimuth direction at HV may appear. Could the effect of these noise components be estimated or evaluated somehow? At least, this should be mentioned in the manuscript.*

**Reply**:  We do notice evident noise patterns related to subswaths in both HH and HV channel, with the HV channel showing higher noise, which is pointed out by the referee. In this manuscript, we do not correct for the incidence angle, nor the subswath-related noise patterns (see our reply to the previous comment). Their effect on the statistical relationship is not fully studied, which is planned for future work.

For revision, we add the proper statements for Sentinel-1 EW mode image noises to the data description part in Section 2 of the revised manuscript (Sun et al., 2021).

*Sun Y, Li X M. Denoising Sentinel-1 Extra-Wide Mode Cross-Polarization Images Over Sea Ice[J]. IEEE Trans. Geosci. Remote. Sens., 2021, 59(3): 2116-2131.*

*In the figures, e.g. Fig. 2 and 3, the plots do not look line there were a linear dependency. It would be good to test regression with second order terms also and see whether they provide better results. In a linear case the higher order regression coefficients would be close to zero anyway.*

**Reply**: We totally agree that incorporating a nonlinear (i.e., second-order) term in the regression model could potentially improve the fitting results. Below, we show the 2nd-order polynomial regression for two sample segments on April 8th and April 12th ($Fs = a*Sigma0^2 + b*Sigma0 + c$):

[Figure]

For both segments, the results show slightly better fitting (i.e., higher R2) than that in the manuscript. Based on the referee's suggestion, we have included the results with nonlinear fitting to the supplementary, along with necessary revisions in the main manuscript.

However, we show the results with the simple linear regression model for the following reasons:

1. We do not fully understand the relationship between the backscatter and the freeboard, which is dominated by the different scattering mechanisms of various sea ice and its snow cover's conditions. Remarkably in the figure above, the relationship in the FYI is already nonlinear for the survey on 12th. Since these underlying mechanisms are not fully understood, we consider exploiting complex fitting models unnecessary at this point.
2. The linear regression model, although simple, captures the statistically significant relationship between the two. Especially, after binning to Sigma0 (Fig. 4 and 5 of the manuscript), the relationship is highly linear at large spatial scales (i.e., 100m and 200m).
3. Last but not the least, we want to reemphasize that our focus for the freeboard upscaling is the prediction of freeboard distribution, not the point-to-point prediction of the freeboard map. The prediction of freeboard distribution does not rely on the specific form of regression from Sigma0 to freeboard.

*Pay attention to the subfigure labels. The labels are now within the subfigures and in some cases in a colored area making them difficult to see, at least in printed versions. The labels (a, b, ...) should be in a fixed position and preferably on white background.*

**Reply**: These figures have been revised in the manuscript.

*In Figs. 2 and 3 also describe the subfigures b and c in the image captions.*

**Reply**: The descriptions have been added in the revised manuscript. (Fig. 2 and 3)

*On page 5 "Single Product Speckle Filter" is mentioned. It is a part of the SW package but still it there should be a reference to a publication where the filter is described or a description of the filter.*

**Reply**: We have added the detail of the treatment and the necessary citation to the reference. The added sentence is: "*To reduce speckle noise in the SAR images, we applied the Lee-Sigma filter using a sliding window of size 7×7*".

The reference is listed below:
> *Mansourpour M, Rajabi M A, Blais J A R. Effects and performance of speckle noise reduction filters on active radar and SAR images, Proc. Isprs. 2006, 36(1): W41.*

*Do not refer to supplement figures in the manuscript. Include the figures referred in the manuscript.*

**Reply**: It has been revised in the manuscript.

*The "statistical fitting" in Section 3.3.1 is not well described. Please, include a detailed description of the method in the manuscript. Have e.g. EM algorithm been used in the fitting?*

**Reply**: We have further described these details in the added "*Methodology*" section.

*What do the Fs estimation would look for the HV data?*

*Would the estimates be better if HH and HV channels were combined in the Fs distribution estimation.*

**Reply**: In our response to the next comment, we show the freeboard map predictions of the entire OIB track with either HH or the HV channel.

Below we show the linear regression between HH backscatter and HV backscatter. The results indicate the RMSE of approximately 1.2 dB and the correlation (Pearson's r) of around 0.9 (see figure below). This suggests that while there is a strong correlation between HH and HV backscatter, HH-channel backscatter alone cannot fully account for the variability observed in HV channel. This result also implies that using the HV channel may offer additional information that could improve the freeboard prediction. We have not fully explored the potential of using both channels for the prediction. This topic serves as a potential direction for future research.

[Figure]

[Figure]

*Would it also be possible to provide some examples of average ice thickness estimated for whole S-1 SAR scenes, e.g. in the current Section 4 and possibley provide a visual comparison to a available coarser scale operational Fb estimates, (e.g. based on*

*CryoSat-2, ICESat-2, SMOS)? The SAR image(s) could be acquired outside of the time window of the study.*

**Reply**:  As suggested by the referee, we have applied the regression model between fs and backscatter to predict the freeboard around the whole OIB track. Specifically, the regression model of 27km-long segments at 200m-scale is adopted, with 9km overlapping by adjacent segments. The figures below show the region of freeboard map prediction (dashed black box), the Sigma0 map in detail, and the corresponding predicted freeboard map. We refrain from providing sea ice thickness maps, since we have limited knowledge of the snow depth over the sea ice cover. These freeboard maps are also added to the supplementary of the revised manuscript.

For OIB segment on April 8th:

[Figure]

[Figure]

[Figure]

For OIB segment on April 12th:

[Figure]

[Figure]

*It would be preferable to have the information in appendix A and B in the manuscript, e.g. the table B1 should be in Section 2.2. In the table the SAR image names could be replaced by the acquisition times to make the table narrower.*

**Reply**: According to the suggestion (also mentioned previously), we have incorporated the content of Appendix B into the revised manuscript. Additionally, we have retained Appendix A as supplementary material. We have also further refined the format of Table B1 to enhance clarity and readability.

---

## Author Comment (AC3)

**Reply to Referee Comment #3 (RC3)**

The authors would like to thank the referee for the invaluable comments and suggestions. The following are the replies for each point of the comment, together with specific revisions that are made. The original comments are in *red italic* font and listed in paragraphs, with our reply following each paragraph separately. The revisions are also highlighted in the revised manuscript in *red* and marked by **RC3**.

*Review of "On the Statistical Relationship between Sea Ice Freeboard and C-Band Microwave Backscatter – A Study with Sentinel-1 and Operation IceBridge"*

*Summary*

*In this paper, the authors use a combination of data from Operation IceBridge, ICESat-2 and Sentinel-1 to investigate the statistical relationship between the altimetric freeboard and C-band backscatter signature for assessing the feasibility to predict the 2D variability of altimeter freeboards that typically offer limited spatial sampling (in comparison to SAR). The analysis carried out in this paper could be a good contribution to the sea ice remote sensing community. However, the paper in its current state is not ready for publication. I recommend major revisions.*

*Major comments*

> *One of my main concerns is the localized aspect of the statistical relationships. You demonstrate that even locally, replicating freeboard with SAR observations is challenging because of many factors: change in local sea ice conditions, scattering mechanisms. Therefore, I am not convinced of the actual usefulness of the work reported here. A more complete analysis would include looking at coincident data between sentinel-1 and ICESat-2 which would provide more confidence on the feasibility of upscaling ICESat-2 measurements.*

**Reply**: Regarding the referee's concern about the locality of the relationship and the usefulness of the method, we would like to argue that the freeboard prediction can be accurately predicted, including the following aspects:

First, the spatial locality of the statistical relationship is mainly caused by the small-scale variability of the sea ice. As shown in our analysis with the 9km segment lengths for deriving the relationships (Fig. 6 of the original manuscript), the regression model parameters (slope and intercept) show gradual changes along the flight track, and they are similar between the inbound and the outbound track. Furthermore, at 27km segment length, the variability of these parameters are effectively reduced, implying that their variability is caused by the limited representation at 9km segment lengths. Therefore, the local spatial window that is large enough for accumulating freeboard-backscatter samples is necessary so as to ensure the stability of the relationship and overcome its locality during prediction.

Second, temporal locality is also required for both the collocation between freeboard and backscatter measurements and the derivation of the prediction model. The temporal locality is influenced by various physical processes. Events like ice melt or snowfall can alter the snow-ice interface properties, potentially degrading the freeboard-backscatter relationship at synoptic scales. Ice deformation can also impact this relationship by degrading the collocation between the two measurements (Sec. 4.3 of the manuscript).

Our supplementary analysis (Fig. S7 and S8) demonstrates significant correlation between freeboard and backscatter over 14-day windows of sample segment on 2019-Apr-8. For the sample segment on April 12th, between April 5th and 12th, significant sea ice drift and deformation is present for the sea ice cover around the sample segment. Correspondingly, the correlation coefficients between Fs and backscatter witness significant drops. This suggests that when the ice conditions permit, the time window for ICESat-2/Sentinel-1 synergy can potentially be extended to 14 days.

In order to accommodate the locality of the statistical relationship, we will carry out large-scale retrieval by both: (1) deriving the relationship with locally collocated observations by the altimeter and SAR satellite, and (2) carrying out prediction within the periphery of these observations.

Finally, although our focus is the prediction of freeboard distributions at the altimeter's native resolution, we prefer to demonstrate the capability of the method with the freeboard map examples using linear regressions and HH-polarized backscatter maps at 200m scale. The results for the OIB campaign on both April 8th and 12th are shown below. We consider that at 200m scale, the linear regression model is good enough for the production of freeboard maps. In the cross track direction, the freeboard maps cover 50km, much wider than the OIB ATM/IS2 swath. In the future study, we will further our study to quantify the optimal temporal and spatial scales for: (1) the collocation between the altimetric scan and SAR maps, and (2) the prediction of freeboard maps and freeboard distributions.

[Figure]

[Figure]

*I think the clarity of the paper could be greatly improved if the structure was revised, and some unnecessary text was removed. A lot of relevant information is in the supplement section and should be moved to the main text. In addition, some technical terms are not defined and datasets not introduced which hinder the understanding of the paper.*

**Reply**: We appreciate your suggestions and have made several revisions to improve the paper structure and enhance its readability.

We have added a new section titled "*Methodology*" immediately after Section 2. This section provides a detailed description of the methods and technical detail used in our study. Moverover, we have moved the content of Appendix B into the main text. We have reviewed the manuscript to ensure that all technical terms are clearly defined and all datasets are adequately introduced.

*Minor comments/questions*

*In the introduction, specifically in the problems section, you talk mostly about the limited synergy between ICESat-2 and SMOS, due to the sampling of ICESat-2. You raise that as a problem you want to address, but it is not mentioned again in the paper.*

**Reply**: The topic of ICESat-2 and SMOS synergy is a related topic in terms of altimetry representation. Indeed we are not able to return to this issue in this study, so we have deleted it in the revised manuscript.

*Why do you collocate OIB passes from the same day? There is an overlap but do you determine the correction from that overlap and apply it to all the cross-track samples?*

**Reply**: The collocation between OIB passes is necessary: since there are both slight sea ice drift and locating uncertainty, the OIB passes can not be merged directly. The main purpose is to merge the passes and form a freeboard map that spans a wider area in the cross-track direction (~1500m). The merged freeboard map facilitates the analysis of its relationship with the backscatter at a range of spatial scales.

We confirm that the overlap of adjacent OIB passes are carried out by their overlapping part. The ATM swath width is about 500m and the adjacent OIB passes overlap by about 10%. And the collocation is based on maximizing the correlation of the freeboard between the adjacent passes (detailed examples in Appendix A). Specifically, we first divide the OIB track in the along-track direction into small segments, and then carry out the collocation and merging of the freeboard map.

The OIB campaigns on April 8th and 12th were organized as racetracks, so for both the outbound and the inbound passes, the merged freeboard map is about 1500m wide. Furthermore the inbound and outbound freeboard maps are separated by about 3000m (i.e., the same separate between beam #1 and #2 of ICESat-2).

*The description of the OIB processing is confusing: you look at the correlation between successive OIB passes on the same day. What does that tell you?*

**Reply**: The correlation is carried out over the overlapping part of adjacent OIB passes. We carry out drift correction to maximize the correlation, in order to merge these passes into a wide freeboard map (examples in Fig. 2 and 3). The correction to maximize the correlation is potentially indicative of the relative drift of the sea ice between the visit OIB times.

For revision and clarity, we add the detail of the correlation between OIB passes before the collocation collocation in Fig. A1 and A2. For the OIB segment on April 12th, the correlation between OIB passes before collocation is indeed low. This is primarily due to the sea ice drift during the different visit times of the OIB passes.

*You need to provide more details on the collocation between S-1 and OIB. Your correlation analysis shows values in the along-track direction only (Fig 4 and 5). But OIB provides you maps of freeboard. Do you apply the same drift correction for all the cross-track samples*

**Reply**: We divided the OIB tracks into 9km segments when collocating with S-1 data. We then performed collocation for each 9km outbound segment and each 9km inbound segment independently. Collocation is performed in both the along-track and cross-track directions.

In Figures 4a and 5a, we showed the drift corrections applied to each segment. Each vector represents the drift correction for a specific 9km segment, including along- and cross-track components. The start position of the vector indicates the original position of the OIB segment before drift correction, while the end position of the vector shows the corrected position after applying the drift correction. Specifically, blue vectors represent the drift corrections applied to the outbound segments, and red vectors represent the drift corrections applied to the inbound segments.

As suggested by the referee, we have added a detailed description of the collocation process between OIB Fs and S-1 in the revised manuscript.

*You mention that you "validate" your OIB freeboard estimates with the OIB Level 4 product. However, this data product is not described in the text. Does it use the same approach for retrieving freeboard?*

**Reply**: We have added an introduction to the official OIB L4 product in Section 2.1.

*How do you determine the reference sea surface elevation for the calculation of the OIB freeboards?*

**Reply**: We adopted the lowest elevation method to retrieve the OIB freeboard. Specifically, we extracted the lowest 1‰ of elevation samples within each 10 km segment and then interpolated these extracted samples to construct the local water level by the Inverse Distance Weighting (IDW) method. The method for determining the sea surface is described in Appendix A. This approach is verified, to first order, through the consistency with the official OIB L4 product.

*Please define interquartiles. Also are you referring to interquartile range?*

**Reply**: We confirm that we are indeed referring to the interquartile range (IQR). As suggested, we have added a definition of the interquartile range to the revised manuscript.

*Line 193: please explain what you mean by:" After binning to $\sigma 0$,"*

**Reply**: The phrase "After binning to $\sigma 0$" refers to the process of grouping the OIB freeboard samples into $\sigma 0$ bins. Specifically, the binning is based on 1 dB increments of $\sigma 0$. Within each bin, we calculated the mean freeboard within the interquartile range. We have added a detailed explanation of this process when the term first appears in the manuscript.

*Line 228: you should define the term "heteroskedasticity"*

**Reply**: We have now avoided the use of the term "heteroskedasticity". It describes the statistical phenomenon where, for larger σ0 bins, both the mean value of freeboard and its variability increase.

*You should define the Kolmogorov-Smirnov (K-S) distance*

**Reply**: We have added the formal definition of the Kolmogorov-Smirnov (K-S) distance to the manuscript.

The K-S distance is a statistical measure used to compare the similarity between two probability distributions, or to assess how well a sample fits a theoretical distribution. When comparing two sample-based distributions, the K-S distance is:

$$D = \sup_x |F_n(x) - G_m(x)|$$

where Fn(x) and Gm(x) are the empirical CDFs of the two samples. The sup (supremum) means the maximum value over all x.

*I believe your description of the ICESat-2 data products is not accurate. Please check the along-track resolution for weak and strong beams as well as the footprint. Also you mention considerable uncertainties (line 418). Please clarify what you mean.*

**Reply**: We have made revisions to be more clear on the ICESat-2 data. For clarification: the *considerable uncertainty* is the per-photon height uncertainty, not the typical ATL07 beam segment height. The ATL07 beam segment height is shown to be lower than 3 centimeters, which is highly precise. The sentence has been revised in the revised manuscript: "*The photon-level elevation measurements represent a similarly fine spatial scale to the OIB ATM, but contain higher uncertainty than that of the beam segment elevation product (ATL07)*".

*Line 316: Your comment on ICESat does not seem to belong here.*

**Reply**: It has been deleted in the revised manuscript.

*Your summary and outlook section is too short. I believe you should merge it with the discussion section.*

**Reply**: According to the suggestion, we have merged the Summary and Outlook section with the Discussion section in the revised manuscript.

---

## Author Response (AR2)

According to the revision suggestions, we have made the following corrections to the manuscript. We sincerely thank the editor and reviewer for the effort for improving the manuscript.

I thank the authors for addressing most of my comments. I believe the manuscript is now in better shape. I have some remaining minor suggestions/comments:

• The description of the algorithm steps should be in the imperative not the present tense (see Section 3.2)

**Reply**: It has been revised in the manuscript.

• I see that the title has been revised to include "case study" which is accurate. However, "case" study does not appear again in the text. I believe this should be explicitly mentioned again, at least in the discussion section.

**Reply**: We have incorporated the term 'case study' in the Outlook section as suggested.

• As suggested, the authors have merged the Summary and Outlook section with the Discussion section. I think the reorganization is good, but the contents of the new "Discussions and summary" section should be trimmed and made more specific. A summary is not a repetition of previous sections; it should be more concise. For example, the paragraph starting with "It is important to note that in this study we did not apply IA corrections to the SAR images." on page 20 does not belong in the summary and discussion section. This is a methodological choice and should be described in the earlier sections. The "Discussions and Summary" section should instead focus on the main results and include a discussion on their validity.

**Reply**: We agree that the 'Discussion and Summary' section required trimming, and that methodological details regarding IA correction belong in earlier sections. Accordingly, we have relocated the IA correction description to Section 2.2.

• In the discussion and summary section (page 24), the authors mention: 'For future work, we plan to further explore the freeboard-backscatter relationship under various conditions.' This style is more appropriate for a proposal than for a scientific paper. It would be better to highlight the limitations of this work (as it is a case study) and suggest that further studies (not necessarily by the authors) should explore the freeboard-backscatter relationship using a larger dataset.

**Reply**: Thanks for the helpful comments. In the revised manuscript in the Discussions and summary section, we have revised this sentence as follows:

"Given the limitation of this case study, future work should explore the freeboard-backscatter relationship under various conditions using larger datasets."

• Page 25: "The photon-level elevation measurements represent a similarly fine spatial scale to the OIB ATM, but contain higher uncertainty than that of the beam segment elevations (ATL07)." Please clarify this statement. ATL07 is derived from the photon product ATL03, and many researchers have used ATL03 over sea ice, although it is more challenging. If you chose not to use ATL03 because of its difficulty, you should clarify that, or else omit the comparison. As written, the statement is misleading and suggests that photon heights are not accurate. At the very least, you should provide a reference to support this claim.

**Reply:** It has been deleted in the revised manuscript.